# Neural Approximation of Graph Topological Features

**Zuoyu Yan**
Wangxuan Institute of Computer Technology
Peking University
yanzuoyu3@pku.edu.cn

**Tengfei Ma**
IBM T. J. Watson Research Center
tengfei.ma1@ibm.com

**Liangcai Gao**
Wangxuan Institute of Computer Technology
Peking University
glc@pku.edu.cn

**Zhi Tang**
Wangxuan Institute of Computer Technology
Peking University
tangzhi@pku.edu.cn

**Yusu Wang**
Halıcıoğlu Data Science Institute
University of California
yusuwang@ucsd.edu

**Chao Chen**[*]
Department of Biomedical Informatics
Stony Brook University
chao.chen.1@stonybrook.edu

## Abstract

Topological features based on persistent homology can capture high-order structural information which can then be used to augment graph neural network methods. However, computing extended persistent homology summaries remains slow for large and dense graphs and can be a serious bottleneck for the learning pipeline. Inspired by recent success in neural algorithmic reasoning, we propose a novel graph neural network to estimate extended persistence diagrams (EPDs) on graphs efficiently. Our model is built on algorithmic insights, and benefits from better supervision and closer alignment with the EPD computation algorithm. We validate our method with convincing empirical results on approximating EPDs and downstream graph representation learning tasks. Our method is also efficient; on large and dense graphs, we accelerate the computation by nearly 100 times.

## 1 Introduction

Graph neural networks (GNNs) have been widely used in various domains with graph-structured data [45, 27, 22, 42, 6]. Much effort has been made to understand and to improve graph representation power [47, 30, 3, 28]. An intuitive solution is to explicitly inject high order information, such as graph topological/structural information, into the GNN models [51, 26]. To this end, persistent homology [15, 14], which captures topological structures (e.g., connected components and loops) and encodes them in a summary called *persistence diagram (PD)*, have attracted the attention of researchers. Indeed, persistence has already been injected to machine learning pipelines for various graph learning tasks [55, 56, 16, 4, 8, 50]. In particular, it has been found helpful to use the so-called *extended persistence diagrams (EPDs)* [10], which contain richer information than the standard PDs.

Despite the usefulness of PDs and EPDs, their computation remains a bottleneck in graph learning. In situations such as node classification [56] or link prediction [50], one has to compute EPDs on vicinity graphs (local subgraph motifs) generated around all the nodes or all possible edges in the input graph. This can be computationally prohibitive for large and dense graphs. Take the Amazon

---
[*]Correspondence to Chao Chen, Yusu Wang, and Liangcai Gao

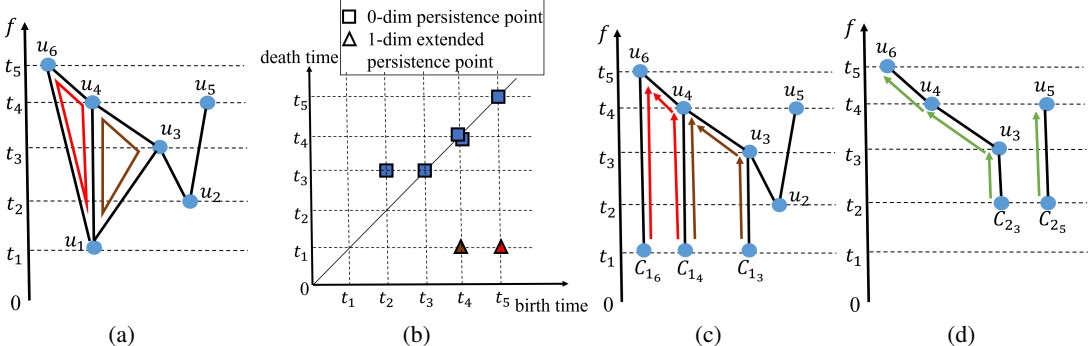

Figure 1: An explanation of extended persistent homology and its computation. The height-function-based filtration is only for illustration purposes. (a) The input graph is plotted with a given filter function. (b) the extended persistence diagram of (a). Commonly speaking, the persistence points on the diagonal (uncritical points) should not be plotted. We plot these points for a clearer illustration. (c) and (d) are examples of finding the loops in the input graph.

Computers dataset [37] as an example. To compute EPDs on vicinity graphs take several seconds on average, and there are 13381 nodes. So to compute all EPDs with a single CPU can take up to a day. This is not surprising as, while theoretically EPD for graphs can be computed in $O(n \log n)$ time [2], that algorithm has not been implemented, and practical algorithms for computing PD take quadratic time in worst case [50].

These computational difficulties raise the question: *can we approximate the expensive computation of EPDs using an efficient learning-based approach?* This is a challenging question due to the complex mathematical machinery behind the original algorithm. First, the algorithm involves a reduction algorithm of the graph incidence matrix. Each step of the algorithm is a modulo-2 addition of columns that can involve edges and nodes far apart. Such algorithm can be hard to be approximated by a direct application of the black-box deep neural networks.

The second challenge comes from the supervision. The output EPD is a point set with an unknown cardinality. The distance between EPDs, called the *Wasserstein distance* [9, 11], involves a complex point matching algorithm. It is nontrivial to design a deep neural network with variable output and to support supervision via such Wassserstein distance. Previous attempts [38, 29] directly use black-box neural networks to generate fixed-length vectorization of the PDs/EPDs and use mean squared error or cross-entropy loss for supervision. The compromise in supervision and the lack of control make it hard to achieve high-quality approximation of PDs/EPDs.

In this paper, we propose a novel learning approach to approximate EPDs on graphs. Unlike previous attempts, we address the aforementioned challenges through a carefully designed learning framework guided by several insights into the EPD computation algorithm.

In terms of model output and supervision, we observe that the computation of EPDs can be treated as an edge-wise prediction instead of a whole-graph prediction. Each edge in the graph is paired with another graph element (either vertex or edge), and the function values of the pair are the coordinates of a persistence point in the EPD. This observation allows us to compute EPDs by predicting the paired element for every edge of the graph. The Wasserstein distance can be naturally decomposed into supervision loss for each edge. This element-wise supervision can significantly improve learning efficiency compared with previous solutions, which treat PDs/EPDs as a whole-graph representation and have to use whole-graph representation pooling.

Another concern is whether and how a deep neural network can approximate the sophisticated EPD algorithm. To this end, we redesign the algorithm so that it is better aligned with algorithms that are known to be learnable by neural networks. Recall we observe that computing EPDs can be decomposed into finding pairing for each edge. We show that the decomposition is not only at the output level, but also at the algorithm level. The complex standard EPD computation algorithm can indeed be decomposed into independent pairing problems, each of which can be solved exactly using a classic *Union-Find algorithm* [12]. To this end, we draw inspiration from recent observations that neural networks can imitate certain categories of sequential algorithms on graphs [43, 46]. We

propose a carefully designed graph neural network with specific message passing and aggregation mechanism to imitate the Union-Find algorithm.

Decomposing the algorithm into Union-Find subroutines and approximating them with a customized GNN provide better alignment between our neural network and the EPD algorithm. A better alignment can lead to better performance [48]. Empirically, we validate our method by quantifying its approximation quality of the EPDs. On two downstream graph learning tasks, node classification and link prediction, we also show that our neural approximations are as effective as the original EPDs. Meanwhile, on large and dense graphs, our method is much faster than direct computation. In other words, the approximated EPDs do not lose accuracy and learning power, but can be computed much more efficiently. Finally, we observe that our model can be potentially transferred to unseen graphs, perhaps due to the close imitation of the Union-Find subroutine. This is encouraging as we may generalize topological computation to various challenging real-world graphs without much additional effort.

In summary, we propose an effective learning approach to approximate EPDs with better supervision and better transparency. The technical contributions are as follows.

- We reformulate the EPD computation as an edge-wise prediction problem, allowing better supervision and more efficient representation learning. We show that the EPD computation can be decomposed into independent pairing problems, each of which can be solved by the Union-Find algorithm.
- Inspired by recent neural algorithm approximation works [43, 46], we design a novel graph neural network architecture to learn the Union-Find algorithm. The closer algorithmic alignment ensures high approximation quality and transferability.

## 2   Background: Extended Persistent Homology

We briefly introduce extended persistent homology and refer the readers to [10, 14] for more details.

**Ordinary Persistent Homology.** Persistent homology captures 0-dimensional (connected components), 1-dimensional (loops) topological structures, as well as high-dimensional analogs, and measures their saliency via a scalar function called *filter function*. Here we will only describe it for the graph setting. Given a input graph $G = (V, E)$, with node set $V$ and edge set $E$, we call all the nodes and edges *simplices*. Denote by $X = V \cup E$ the set of all simplices. We define a filter function on all simpices, $f : X \to \mathbb{R}$. In the typical sublevel-set setting, $f$ is induced by a node-valued function (e.g., node degrees), and further defined on edges as $f(uv) = max(f(u), f(v))$.

Denote by $X_a$ the *sublevel set of* $X$, consisting of simplices whose filter function values $\leq a$, $X_a = \{x \in X | f(x) \leq a\}$. As the threshold value $a$ increases from $-\infty$ to $\infty$, we obtain a sequence of growing spaces, called an *ascending filtration* of $X$: $\emptyset = X_{-\infty} \subset ... \subset X_\infty = X$. As $X_a$ increases from $\emptyset$ to $X$, new topological structures gradually appear (born) and disappear (die). For instance, the blue square persistence point at $(t_2, t_3)$ in Figure 1 (b) indicates that the connected component $u_2$ appears at $X_{t_2}$ and is merged with the whole connected component at $X_{t_3}$.

Applying the homology functor to the filtration, we can more precisely quantify the birth and death of topological features (as captured by homology groups) throughout the filtration, and the output is the so-called *persistence diagram (PD)*, which is a planar multiset of points, each of which $(b, d)$ corresponds to the birth and death time of some homological feature (i.e., components, loops, and their higher dimensional analogs). The lifetime $|d - b|$ is called the *persistence* of this feature and intuitively measures its importance w.r.t. the input filtration.

**Extended Persistent Homology.** In the ordinary persistent homology, topology of the domain (e.g., the graph) will be created at some time (has a birth time), but never dies (i.e., with death time being equal to $+\infty$). We call such topological features *essential features*. In the context of graphs, the importance of 1D essential features, corresponding to independent loops, are not captured via the ordinary persistence. To this end, an *extended persistence module* is introduced in [10]: $\emptyset = H(X_{-\infty}) \to \cdots H(X_a) \to \cdots H(X) = H(X, X^\infty) \to \cdots \to H(X, X^a) \to \cdots \to H(X, X^{-\infty})$, where $X^a = \{x \in X | f(x) \geq a\}$ is a *superlevel set* of $X$ at value $a$. We say that the second part $H(X, X^\infty) \to \cdots \to H(X, X^a) \to \cdots \to H(X, X^{-\infty})$ is induced by a *descending filtration*. If we inspect the persistence diagram induced by this extended sequence, as $H(X, X^{-\infty})$ is trivial, all the loop features created will also be killed in the end, and thus captured by persistence points whose

birth happens in the ascending filtration and death happens in the descending filtration. In what follows, we abuse the notation slightly and use *1D EPD* to refer to only such persistence points (i.e., born in ascending portion and death in descending portion) in the persistence diagram induced by the extended module[2]. We use 0D PD to refer to the standard ordinary 0D persistence diagram induced by the ascending sequence. Our goal is to compute/approximate the union of 0D PD and 1D EPD.

Specifically, in the graph setting, at the end of the ascending filtration, some edges, which are the so-called negative edges (as they kill homological features), are paired with the vertices. These correspond to points in the 0D PD, capturing the birth and death of connected components in the ascending filtration. Those unpaired edges, called *positive edges*, will create independent loops (1D homology for graphs) and remain unpaired after the ascending filtration. The number of such unpaired edges equals to the 1st Betti number $\beta_1$ (rank of the 1st homology group). These edges will then be paired in the descending part of the persistence module and their birth-depth times give rise to 1D EPD. An example is given in Figure 1(b). Note that since our domain is a graph, $\beta_1$ equals the number of independent loops, which also equals to $\beta_1 = |E| - |V| + 1$ for a connected graph. Hence we also say that 1D EPD captures the birth and death of independent loop features. The birth and death times of the loop feature correspond to the threshold value $a$'s when these events happen. In general, the death time for such loop feature is smaller than the birth time. For example, the red triangle persistence point in Figure 1 (b) denotes that the red cycle in Figure 1 (a) appears at $X_{t_5}$ in the ascending filtration and appears again at $X^{t_1}$ in the descending filtration.

Finally, PDs live in an infinite-dimensional space equipped with an appropriate metric structure, such as the so-called $p$-th Wasserstein distance [11] or the bottleneck distance [9]. They have been combined with various deep learning methods including kernel machines [34, 25, 5], convolutional neural networks [18, 20, 44, 57], transformers [53], connectivity loss [7, 17], and GNNs [56, 8, 50, 55, 16, 4]. During learning, there have been many works in the literature to vectorize persistence diagrams for downstream analysis. Among these works a popular choice is the persistence image [1].

# 3 Algorithm Revision: Decomposing EPD into Edge-Wise Paring Predictions

In this section, we provide algorithmic insights into how the expensive and complex computation of EPDs can be decomposed into pairing problems for edges. And each pairing problem can be solved exactly using a Union-Find algorithm. The benefit is two-folds. First, the decomposition makes it possible to train the neural network through edge-wise supervision.This allows us to adopt the popular and effective edge-prediction GNN for the goal. Second, we observe the similarity between the Union-Find and sequential algorithms which are known to be imitable by neural networks. This gives us the opportunity to design a special graph neural network to imitate the algorithm accurately, and to approximate EPDs accurately.

**Decompose the EPD Computation into Pairing Computations.** Recall that our goal is to compute the 0D PDs and 1D EPDs $PD_0$ and $PD_1$. The reason for not estimating 0D EPDs (or not including the global max/min pair that corresponds to the whole connected component) is that (1) the global max/min value is easy to obtain, and does not need an extra prediction; (2) in our setting, the global max/min pair will not be paired with any edge in the ascending filtration. In the later section, the estimation of EPDs denote the estimation of $PD_0$ and $PD_1$.

We observe that on these diagrams, each point corresponds to a unique pairing of graph elements (vertex-edge pair for $PD_0$, edge-edge pair for $PD_1$). Each pair of elements are essentially the "creator" and "destroyer" of the corresponding topological feature during the filtration. And their filtration values are the birth and death times of the topological feature. For example, the persistence point located at $(t_2, t_3)$ in Figure 1 (b) denotes that the edge $u_2u_3$ is paired with $u_2$. We consider the following "unique pairing" for all edges in the graph: Consider each edge in the ascending filtration: if the edge is a destroyer in the ascending filtration, it will be paired with a vertex. Otherwise, this edge $e$ is a creator in the ascending filtration and will be paired during the descending filtration with another edge $e'$. We note that this is not in conflict with the fact that the PDs/EPDs are often sparse. Many pairings are local and only pair adjacent elements. They correspond to zero-persistence points living in the diagonal of the diagrams.

---

[2]We note that in standard terminology, extended persistence diagram will also contain persistent points born and destroyed both in the descending sequence.

**Algorithm 1** Sequential algorithm

1: **Input:** graph $G = (V, E)$, filter function $f$.
2: Initialise-Nodes$(V, f)$
3: $Q = $ Sort-Queue$(V)$
4: **while** Q is not empty **do**
5:     $u = Q$.pop-min()
6:     **for** $v \in G$.neighbors$(u)$ **do**
7:       Relax-Edge$(u, v, f)$
8:     **end for**
9: **end while**

---

**Algorithm 2** Computation of EPD

1: **Input:** filter function $f$, input graph $G = (V, E)$
2: $V, E = $ sorted$(V, E, f)$
3: $PD_0 = $ Union-Find$(V, E, f)$, $PD_1 = \{\}$
4: **for** $i \in V$ **do**
5:     $C_i = \{C_{i_j} | (i, j) \in E, f(j) > f(i)\}$, $E_i = E$
6:     **for** $C_{i_j} \in C_i$ **do**
7:       $f(C_{i_j}) = f(i), E_i = E_i - \{(i, j)\} + \{(C_{i_j}, j)\}$
8:     **end for**
9:     $PD_1^i = $ Union-Find-step$(V + C_i - \{i\}, E_i, f, C_i)$
10:     $PD_1 += PD_1^i$
11: **end for**
12: **Output:** $PD_0, PD_1$

---

**Algorithm 3** Union-Find-step (Sequential)

1: **Input:** $V, E, f, C_i$
2: $PD_1^i = \{\}$
3: **for** $v \in V$ **do**
4:     $v.value = f(v), v.root = v$
5: **end for**
6: $Q = $ Sort$(V), Q = Q - \{v | f(v) < f(i)\}$, $G = \{Q, E_Q\}$, where $E_Q = E \cup Q^2$.
7: **while** Q is not empty **do**
8:     $u = Q$.pop-min()
9:     **for** $v \in G$.neighbors$(u)$ **do**
10:
11:       $pu, pv = $ Find-Root$(u)$, Find-Root$(v)$
12:       **if** $pu \neq pv$ **then**
13:         $s = argmin(pu.value, pv.value)$
14:         $l = argmax(pu.value, pv.value)$
15:         $l.root = s$
16:         **if** $pu \in C_i$ and $pv \in C_i$ **then**
17:           $PD_1^i + \{(u.value, l.value)\}$
18:         **end if**
19:       **end if**
20:     **end for**
21: **end while**
22: **Function:** Find-Root$(u)$
23: $pu = u$
24: **while** $pu \neq pu.root$ **do**
25:     $pu.root = (pu.root).root, pu = pu.root$
26: **end while**
27: **Return:** $pu$

---

This pairing view gives us the opportunity to transform the computation of EPDs into a pairing prediction problem: for every edge in the graph, we predict its pairing element. This will be the foundation of our design of the GNN in Sec. 4. Meanwhile, we observe that *the decomposition is not only at the output level*. The original algorithm of EPD, a sequential modulo-2 matrix reduction algorithm, can indeed be rewritten into a set of independent algorithm subroutines, each for the computation of one pairing. Each subroutine is a Union-Find algorithm. This new decomposed EPD algorithm has not been reported before, although the idea follows from existing work [2]. For completeness, we will provide a proof of correctness of the algorithm.

**Description of Algorithm 2.** The pseudocode for 1D EPD computation is shown in Algorithm 2. We leave the algorithm for 0D PD to the supplementary material[3]. For simplicity of presentation, we assume that all vertices have distinct function values $f : V \to \mathbb{R}$[4]. Therefore finding the persistence value equals to finding the pairing. To compute the EPD, we traverse all nodes in the vertex set and find their extended persistence pairing. Combining the persistence pair from all nodes, we can obtain the final EPD. The algorithm complexity analysis is provided in the supplementary material.

**Finding persistence pairing for nodes.** For node $u_i \in V$, we can call Algorithm 3 to identify the corresponding persistence pair. In particular, the algorithm first sorts the graph elements according to an input scalar function, then does the edge operation by finding the roots of the corresponding nodes and merging these nodes. See Figure 1(c) for a simple illustration. For node $u_1$, there are three upper edges: $u_1 u_3$, $u_1 u_4$, and $u_1 u_6$. We put each such edge $u_i u_j$ in a different component $C_{i_j}$, – we call this *upper-edge splitting operation* – and start to sweep the graph in increasing values starting at $f(u_i)$. Then, the first time any two such components merge will give rise to a new persistence

---

[3]The 0D algorithm needs a single run of Union-Find [14, 13], and is very similar to Algorithm 3 which is a subroutine used by Algorithm 2.

[4]We can add jitter to the original filter function. The output EPDs will only have minor changes [9]

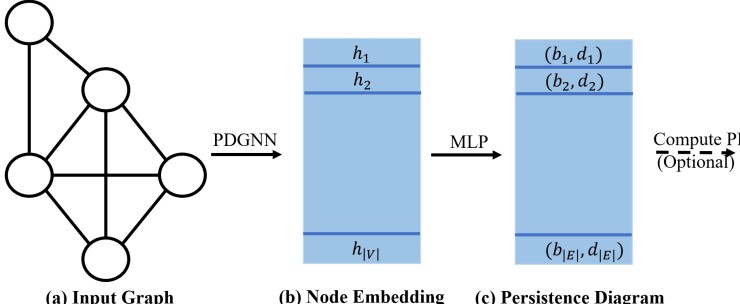

(a) Input Graph     (b) Node Embedding    (c) Persistence Diagram

Figure 2: The basic framework.

point in the 1D EPD. For instance, $C_{1_4}$ and $C_{1_3}$ first merge at $u_4$, and this will give rise to the brown loop in Figure 1(a) with $(t_4, t_1)$ as its persistence point. While in Figure 1 (d), the two connected components, $C_{2_3}$ and $C_{2_5}$ (originated from $u_2$) will not be united. Therefore, node $u_2$ will not lead to any persistence point in the EPD.

**Correctness.** The idea behind Algorithm 2 to compute the extended pairing for essential edges appears to be folklore. For completeness, we provide a proof of its correctness (stated in Theorem 3.1). We provide a sketch of the proof here, leaving the complete proof to the supplementary material.

**Theorem 3.1.** *Algorithm 2 outputs the same 1D EPDs as the standard EPD computation algorithm.*

*Proof sketch.* To compute the 1D EPDs, we simply need to find the pairing partner for all edges. Therefore, to prove that the two algorithms output the same 1D EPDs, we need to prove that the output pairing partners are the same (or share the same filter value). We prove this by showing that both the standard EPD computation algorithm and Algorithm 2 find the "thinnest pair", i.e., the paired saddle points are with the minimum distance in terms of filter value, for all edges.

**Neural Approximation of Union-Find.** In the previous paragraph, we showed that the computation of 1D EPDs can be decomposed into the parallel execution of Union-Find algorithms, which share a similar sequential behavior. This gives us the opportunity to approximate these Union-Find algorithms well, and consequently approximate EPDs well.

Approximating algorithms with neural networks is a very active research direction [52, 21, 24, 33, 35, 49]. Within the context of graph, GNNs have been proposed to approximate parallel algorithms (e.g., Breadth-First-Search) and sequential algorithms (e.g., Dijkstra) [43, 41, 46]. Particularly relevant to us is the success in approximating the category of sequential algorithms such as Dijkstra. These sequential algorithms, as generally defined in Algorithm 1, sort graph elements (vertices and edges) according to certain function, and perform algorithmic operations according to the order. As described in previous paragraphs, the Union-Find algorithm also contains these steps, and can be expressed in a sequential-like form (Algorithm 3). Therefore we propose a framework to simulate the algorithm.

## 4   A Graph Neural Network for EPD Approximation

Previous section establishes the algorithm foundation by showing that we can decompose EPD computation into edge pairing prediction problems, each of which can be solved using a Union-Find algorithm. Based on such algorithmic insights, we next introduce our neural network architecture to approximate the EPDs on graphs. Our main contributions are: (1) we transform the EPD computation into an edge-wise prediction problem, and solve it using a GNN framework, inspired by the GNN for link prediction; (2) we design a new backbone GNN model **PDGNN** to approximate the Union-Find algorithm, with specially designed pooling and message passing operations.

### 4.1   EPD computation as a edge-wise prediction problem

We have established that computing $PD_0$ and $PD_1$ can be reduced into finding the pairing partners for all edges. We transfer the problem into an edge-wise prediction problem. We predict the persistence pairing for all edges. This is very similar to a standard link prediction problem [6, 50], in which one predicts for each node pair of interest whether it is a real edge of the graph or not.

Inspired by standard link-prediction GNN architectures [6, 50], we propose our model (see Figure 2) as follows. (1) For an input graph $G = (V, E)$ and a filter function $f$, we first obtain the initial filter value for all the nodes: $X = f(V) \in R^{|V|*1}$, and then use a specially designed GNN model which later we call PDGNN $\mathcal{G}$ to obtain the node embedding for all these vertices: $H = \mathcal{G}(X) \in R^{|V|*d_H}$. (2) Subsequently, a MLP (Multi-layer perceptron) $W$ is applied to the node embeddings to obtain a two dimensional output for each edge $(u, v) \in E$, corresponding to its persistence pairing. Formally, we use $PP_{uv} = W([h_u \bigoplus h_v]) \in R^2$ as the persistence pair. Here, $h_u$ and $h_v$ denote the node embedding for node $u$ and $v$, and $\bigoplus$ represents the concatenation of vectors.

In Algorithm 2, the Union-Find-step should be implemented on all edges to obtain 1D EPDs. Hence ideally we would need a large GNN model with node features proportional to the graph size so as to simulate all these Union-find-steps in parallel simultaneously. However this would be expensive in practice. On the other hand, there are many overlapping or similar computational steps between the Union-Find-step procedures on different vertices. Hence in practice, we only use bounded-size node features.

## 4.2 PDGNN

In this section, we explain how to design the backbone GNN to approximate the Union-Find algorithm. Note the Union-Find is similar to known sequential algorithms but with a few exceptions. We design specific pooling and message passing operations to imitate these special changes. These design choices will be shown to be necessary in the experiment section.

Recall a typical GNN learns the node embedding via an iterative aggregation of local graph neighbors. Following [47], we write the $k$-th iteration (the $k$-th GNN layer) as:

$$h_u^k = AGG^k(\{MSG^k(h_v^{k-1}), v \in N(u)\}, h_u^{k-1}) \tag{1}$$

where $h_u^k$ is the node features for node $u$ after $k$-th iterations, and $N(u)$ is the neighborhood of node $u$. In our setting, $h_u^0 = x_u$ is initialized to be the filter value of node $u$. Different GNNs have different $MSG$ and $AGG$ functions, e.g., in GIN [47], the message function $MSG$ is a MLP followed by an activation function, and the aggregation function $AGG$ is a sum aggregation function.

We now describe our specially designed GNN, called **PDGNN** (Persistence Diagram Graph Neural Network). Compared with the Sequential algorithms (Algorithm 1) [46], our Union-Find algorithm (Algorithm 3) differs in: (1) the Find-Root algorithm which needs to return the minimum of the component, (2) additional edge operations such as upper-edge splitting. To handle these special algorithmic needs, our PDGNN modifies standard GNNs with the following modules.

**A new aggregation due to the Find-Root function.** Finding the minimum intuitively suggests using a combination of several local min-aggregations. Considering that the sum aggregation can bring the best expressiveness to GNNs [47], we implement the root-finding process by a concatenation of sum aggregation and min aggregation as our aggregation function. To be specific:

$$AGG^k(.) = SUM(.) \bigoplus MIN(.) \tag{2}$$

**Improved edge operations.** As shown in [43, 46], classic GNNs are not effective in "executing" Relax-Edge subroutines. Furthermore, in Algorithm 2, we also need the upper-edge splitting operation for each vertex. In other words, the information of the separated components $C_{i_j}$ are formed by the information from both nodes $u_i$ and $u_j$. To this end, we use edge features and attention to provide bias using edges. Specifically, we propose the following message function in the $k$-th iteration:

$$MSG^k(h_v^{k-1}) = \sigma^k[\alpha_{uv}^k(h_u^{k-1} \bigoplus h_v^{k-1})W^k] \tag{3}$$

where $\sigma^k$ is an activation function, $W^k$ is a MLP module, and $\alpha_{uv}^k$ is the edge weight for $uv$. We adopt $PRELU$ as our activation function, and the edge weight proposed in [42] as our edge weight.

**Training PDGNN.** We use the 2-Wasserstein distance between the predicted diagram and the ground truth EPD as the loss function. Through optimal matching, the gradient is passed to each predicted persistence pair. Since we have established the one-to-one correspondence between pairs and edges, the gradient is then passed to the corresponding edge, and contributes to the representation learning.

Table 1: Approximation error on different vicinity graphs

| Dataset | Cora | | Citeseer | | PubMed | | Photo | | Computers | |
|---|---|---|---|---|---|---|---|---|---|---|
| Evaluation | $W_2$ | PIE | $W_2$ | PIE | $W_2$ | PIE | $W_2$ | PIE | $W_2$ | PIE |
| GIN_PI | — | 5.03e-1 | — | 2.17e-1 | — | 4.08e-1 | — | 5.53 | — | 2.70 |
| GAT_PI | — | 1.43e-1 | — | 1.95e-1 | — | 1.60 | — | 20.98 | — | 44.50 |
| GAT | 0.655 | 2.46e-2 | 0.431 | 4.04e-2 | 0.697 | 3.5e-1 | 1.116 | 1.09 | 1.145 | 2.21 |
| GAT (+MIN) | 0.579 | 1.53e-2 | 0.344 | 1.02e-2 | 0.482 | 4.60e-2 | 0.820 | 1.35 | 0.834 | 0.64 |
| PDGNN (w/o ew) | 0.692 | 2.77e-2 | 0.397 | 2.24e-2 | 0.666 | 9.01e-2 | 2.375 | 6.47 | 18.63 | 27.35 |
| PDGNN | **0.241** | **4.75e-4** | **0.183** | **4.43e-4** | **0.256** | **8.95e-4** | **0.224** | **4.33e-3** | **0.220** | **6.20e-3** |

Table 2: Classification accuracy on various node classification benchmarks

| Method | Cora | Citeseer | PubMed | Computers | Photo | CS | Physics |
|---|---|---|---|---|---|---|---|
| GCN | 81.5±0.5 | 70.9±0.5 | 79.0±0.3 | 82.6±2.4 | 91.2±1.2 | 91.1±0.5 | 92.8±1.0 |
| GAT | **83.0±0.7** | **72.5±0.7** | 79.0±0.3 | 78.0±19.0 | 85.1±20.3 | 90.5±0.6 | 92.5±0.9 |
| HGCN | 78.0±1.0 | 68.0±0.6 | 76.5±0.6 | 82.1±0.0 | 90.5±0.0 | 90.5 ± 0.0 | 91.3±0.0 |
| PEGN (True Diagram) | 82.7±0.4 | 71.9±0.5 | **79.4±0.7** | 86.6±0.6 | **92.7±0.4** | **93.3±0.3** | **94.3±0.1** |
| PEGN (GIN_PI) | 81.8±0.1 | 65.7±2.1 | 77.7±0.9 | 82.4±0.5 | 88.3±0.7 | 92.6±0.3 | 93.7±0.5 |
| PEGN (PDGNN) | 82.0±0.5 | 70.8±0.5 | 78.7±0.6 | **86.7±0.9** | 92.2±0.2 | 93.2±0.2 | 94.2±0.2 |

# 5 Experiments

In this section, we thoroughly evaluate the proposed model from 3 different perspectives. In Section 5.1, we evaluate the approximation error between the predicted diagram and the original diagram and show that the prediction is very close to the ground truth. Even with a small approximation error, we still need to know how much does the error influence downstream tasks. Therefore, in Section 5.2, we evaluate the learning power of the predicted diagrams through 2 downstream graph representation learning tasks: node classification and link prediction. We observe that the model using the predicted diagrams performs comparably with the model using the ground truth diagrams. In Section 5.3, we evaluate the efficiency of the proposed algorithm. Experiments demonstrate that the proposed method is much faster than the original algorithm, especially on large and dense graphs. Source code is available at https://github.com/pkuyzy/TLC-GNN.

**Datasets.** To compute EPDs, we need to set the input graphs and the filter functions. Existing state-of-the-art models on node classification [56] and link prediction [50] mainly focus on the local topological information of the target node(s). Following their settings, for a given graph $G = (V, E)$, we extract the $k$-hop neighborhoods of all the vertices, and extract $|V|$ vicinity graphs. In our experiments, $k$ is set to 1 or 2 (details are provided in the supplementary material).

In terms of filter functions, we use Ollivier-Ricci curvature [31], heat kernel signature with two temprature values [39, 19] and the node degree[5]. For an input vicinity graph, we compute 4 EPDs based on the 4 filter functions, and then vectorize them to get 4 persistence images [1]. Therefore, we can get $4|V|$ EPDs in total. The input graphs include (1) citation networks including Cora, Citeseer, and PubMed [36]; (2) Amazon shopping datasets including Photo and Computers [37]; (3) coauthor datasets including CS and Physics [37]. Details are available in the supplementary material.

---

[5]Following the settings in [56, 50], we adopt the Ollivier-Ricci curvature as the graph metric, and the distance to target node(s) as the filter function; Following the settings in [4], we set the temparature $t = 10$ and 0.1 and adopt these two kernel functions as the filter functions; Node degree is used as the initial filter function in [16].

Table 3: AUC-ROC score on various link prediction benchmarks

| Method | Cora | Citeseer | PubMed | Photo | Computers |
|---|---|---|---|---|---|
| GCN | 90.5± 0.2 | 82.6±1.9 | 89.6±3.7 | 91.8±0.0 | 87.8±0.0 |
| GAT | 72.8± 0.2 | 74.8±1.5 | 80.3±0.0 | 92.9±0.3 | 86.4±0.0 |
| HGCN | 93.8±0.1 | **96.6±0.1** | 96.3±0.0* | 95.4±0.0 | 93.6±0.0 |
| P-GNN | 74.1±2.4 | 73.9±2.6 | 79.6±0.5 | 90.9±0.7 | 88.3±1.0 |
| SEAL | 91.3±5.7 | 89.8±2.3 | 92.4±1.2 | 97.8±1.3 | 96.8±1.5 |
| TLC-GNN (True Diagram) | 94.9±0.4 | 95.1± 0.7 | **97.0±0.1** | 98.2±0.1 | 97.9±0.1 |
| TLC-GNN (GIN_PI) | 93.5±0.2 | 93.3±0.6 | 96.3 ± 0.2 | 95.8± 1.0 | 96.2±0.3 |
| TLC-GNN (PDGNN) | **95.0±0.3** | 95.6±0.4 | **97.0±0.1** | **98.4±0.6** | **98.2±0.3** |

Table 4: Time evaluation on different datasets (seconds)

| Dataset | Cora | Citeseer | PubMed | Photo | Computers | CS | Physics |
|---|---|---|---|---|---|---|---|
| Avg. N/E | 38/103 | 16/43 | 61/190 | 797/16042 | 1879/47477 | 97/431 | 193/1315 |
| Fast [50] | 0.95 | 0.39 | 2.15 | 362.60 | 1195.66 | 5.72 | 24.14 |
| Gudhi [40] | **0.44** | **0.21** | **1.00** | 583.55 | 8585.50 | **3.00** | 26.58 |
| Ours | 5.21 | 4.72 | 4.78 | **6.67** | **7.32** | 5.18 | **5.42** |

## 5.1 Approximation Quality

In this section, we evaluate the approximation error between the prediction and the original EPDs.

**Evaluation metrics.** Recall that the input of our model is a graph and a filter function, and the output is the predicted EPD. After obtaining the predicted EPD, we vectorize it with persistence image [1] and evaluate (1) the 2-Wasserstein ($W_2$) distance between the predicted diagram and the ground truth EPD; (2) the total square error between the predicted persistence image and the ground truth image (persistence image error, denoted as PIE). Considering that our aim is to estimate EPDs on graphs rather than roughly approximating persistence images, we use the $W_2$ distance as the training loss, while the PIE is only used as an evaluation metric. Given an input graph (e.g., Cora, Citeseer, etc.) and a filter function, we extract the $k$-hop neighborhoods of all the vertices and separate these vicinity graphs randomly into 80%/20% as training/test sets. We report the mean $W_2$ distance between diagrams and PIE on different vicinity graphs and 4 different filter functions.

**Baseline settings.** **PDGNN** denotes our proposed method, that is, the GNN framework with the proposed $AGG$ function and $MSG$ function. Its strategy is to first predict the EPD, and then convert it to the persistence image. To show its superiority, we compare with the strategy from [38, 29], i.e., directly approximate the persistence image of the input graph, as a baseline strategy. **GIN_PI** and **GAT_PI** denote the baseline strategy with GIN [47] and GAT [42] as the backbone GNNs.

To show the effectiveness of the modules proposed in Section 4, we add other baselines with our proposed strategy. **GAT** denotes GAT as the backbone GNN. **GAT (+MIN)** denotes GAT with the new $AGG$ function. Compared with PDGNN, it exploits the original node feature rather than the new edge feature in the $MSG$ function. **PDGNN (w/o ew)** denotes PDGNN without edge weight. Further experimental settings can be found in the supplementary material.

**Results.** Table 1 reports the approximation error, we observe that PDGNN outperforms all the baseline methods among all the datasets. The comparison between GAT and GAT_PI shows the benefit of predicting EPDs instead of predicting the persistence image. Comparing GAT and GAT (+MIN), we observe the advantage of the new $AGG$ function, which shows the necessity of using min aggregation to approximate the Find-Root algorithm; Comparing GAT (+MIN) and PDGNN, we observe the effectiveness of using the new $MSG$ function to help the model capture information of the separated connected components. The comparison between PDGNN (w/o ew) and PDGNN shows that edge weights help the model focus on the individual Relax-Edge sub-algorithm operated on every edge.

## 5.2 Downstream Tasks

In this section, we evaluate the performance of the predicted diagrams on 2 graph representation learning tasks: node classification and link prediction. We replace the ground truth EPDs in state-of-the-art models based on persistence [56, 50] with our predicted diagrams and report the results.

**Baselines.** We compare our method with various state-of-the-art methods. We compare with popular GNN models including **GCN** [22], **GAT** [42] and **HGCN** [6]. For link prediction, we compare with several state-of-the-art methods such as **SEAL** [54] and **P-GNN** [51]. Notice that GCN and GAT are not originally designed for link prediction, therefore we follow the settings in [6, 50], that is, to get the node embedding through these models, and use the Fermi-Dirac decoder [23, 32] to predict whether there is a link between the two target nodes. In comparison with the original EPD, we also add **PEGN** [56] and **TLC-GNN** [50] as baseline methods. Furthermore, to show the benefit of directly predicting EPDs, we also add the baseline methods **PEGN (GIN_PI)** and **TLC-GNN (GIN_PI)**, which replace the original persistent homology feature with the output from GIN_PI.

**Evaluation metrics.** For node classification, our setting is the same as [22, 42, 56]. To be specific, we train the GNNs with 20 nodes from each class and validate (resp. test) the GNN on 500 (resp. 1000) nodes. We run the GNNs on these datasets 10 times and report the average classification accuracy

and standard deviation. For link prediction, our setting is the same as [6, 50]. To be precise, we randomly split existing edges into 85/5/10% for training, validation, and test sets. An equal number of non-existent edges are sampled as negative samples in the training process. We fix the negative validation and test sets, and randomly select the negative training sets in every epoch. We run the GNNs on these datasets 10 times and report the mean average area under the ROC curve (ROCAUC) scores and the standard deviation.

**Results.** Table 2 and Table 3 summarize the performance of all methods on node classification and link prediction. We observe that PEGN (PDGNN) and TLC-GNN (PDGNN) consistently perform comparably with PEGN and TLC-GNN, showing that the EPDs approximated by PDGNN have the same learning power as the true EPDs. Furthermore, PEGN using the approximated EPDs achieve better or comparable performance with different SOTA methods.

We also discover that PEGN (GIN_PI) and TLC-GNN (GIN_PI) perform much inferior to the original models using the true EPDs. It demonstrates that the large approximation error from GIN_PI lose much of the crucial information which is preserved in PDGNN.

**Transferability.** One appealing feature of our method is its transferability. Training on one graph, our algorithm can estimate EPDs well on another graph. This makes it possible to apply the computationally expensive topological features to a wide spectrum of real-world graphs; we can potentially apply a pre-trained model to large and dense graphs, on which direct EPD computation is infeasible. The experiments are provided in the supplementary material.

### 5.3 Algorithm Efficiency

In this section, we evaluate the efficiency of our proposed model. For a fair and complete comparison, we compare with algorithms from Gudhi [40] and from [50]. We select the first 1000 nodes from Cora, Citeseer, PubMed, Photo, Computers, CS, Physics, and then extract their 2-hop neighborhoods. With Ollivier-Ricci curvature as the filter function, we compute the EPDs and report the time (seconds) used to infer these diagrams.

**Results.** We list the average nodes and edges of these vicinity graphs in the first line of Table 4. As shown in Table 4, although our model is slower on small datasets like Cora or Citeseer, it is much faster on large and dense datasets. Therefore we can simply use the original algorithm to compute the EPDs on small graphs, and use our model to estimate EPDs on large graphs. The model can be applied to various graph representation learning works based on persistent homology.

## 6 Conclusion

Inspired by recent success on neural algorithm execution, we propose a novel GNN with different technical contributions to simulate the computation of EPDs on graphs. The network is built on algorithmic insights, and benefits from better supervision and closer alignment with the EPD computation algorithm. Experiments show that our method achieves satisfying approximation quality and learning power while being significantly faster than the original algorithm on large and dense graphs. Another strength of our method is the transferability: training on one graph, our algorithm can still approximate EPDs well on another graph. This makes it possible to apply the computationally expensive topological features to a wide spectrum of real-world graphs.

**Acknowledgements.** We thank all anonymous reviewers for their constructive feedback very much. This work of Zuoyu Yan, Liangcai Gao, and Zhi Tang is supported by the projects of National Key R&D Program of China (2019YFB1406303) and National Natural Science Foundation of China (No. 61876003), which is also a research achievement of Key Laboratory of Science, Technology and Standard in Press Industry (Key Laboratory of Intelligent Press Media Technology).

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
