# Neural Approximation of Graph Topological Features
## – Supplementary Material –

**Zuoyu Yan**
Wangxuan Institute of Computer Technology
Peking University
`yanzuoyu3@pku.edu.cn`

**Tengfei Ma**
IBM T. J. Watson Research Center
`tengfei.ma1@ibm.com`

**Liangcai Gao**
Wangxuan Institute of Computer Technology
Peking University
`glc@pku.edu.cn`

**Zhi Tang**
Wangxuan Institute of Computer Technology
Peking University
`tangzhi@pku.edu.cn`

**Yusu Wang**
Halıcıoğlu Data Science Institute
University of California
`yusuwang@ucsd.edu`

**Chao Chen**[*]
Department of Biomedical Informatics
Stony Brook University
`chao.chen.1@stonybrook.edu`

In the supplementary material, we provide (1) the related work; (2) the complexity and the correctness of the introduced algorithm; (3) the Union-Find algorithm in a sequential format; (4) additional experimental details, including the introduction of the datasets, and the experimental settings; (5) additional experiments, including the evaluation on transferability, the influence of training samples, experiments on other datasets, experiments on other attributes of the model and the limitation of the paper.

## 0.1 Related Works

**Learning with Persistent Homology.** Based on the theory of algebraic topology [29], persistent homology [15, 14] extends the classical notion of homology, and can capture the topological structures (e.g., loops, connected components) of the input data in a robust [7] manner. It has already been used in various deep learning domains including kernel machines [32, 25, 4], convolutional neural networks [19, 22, 40, 49], transformers [46], connectivity loss [5, 18], and graph representation learning [48, 6, 44, 47, 17, 3, 26]. Some following works propose persistence-inspired frameworks on other tasks such as knowledge graph completion [43].

**Neural Algorithm Execution.** Many works have studied neural execution in different domains before [45, 23, 24, 31, 33, 42]. With the rapid development of GNNs in graph representation learning, learning graph algorithms with GNNs has attracted researchers' attention [39, 38, 41]. These works exploit GNNs to approximate certain classes of graph algorithms, such as parallel algorithms (e.g., Breadth-First-Search) and sequential algorithms (e.g., Dijkstra). Although the computation of extended persistence diagrams can be written in a sequential-like form, it needs extra steps and considerations. In our framework, we propose different modules to approximate these steps and achieve satisfying practical performance.

**Accelerating Extended Persistent Homology.** In general, computing extended persistent homology relies on the well-known matrix reduction algorithm [8]. Much effort has been made to accelerate the computation, but it still takes matrix multiplication time [37, 12, 10]. For the specific case where the input is a function on a graph $G = (V, E)$, it turns out that one can compute it in $O(|E| \log |V|)$

---

[*]Correspondence to Chao Chen, Yusu Wang, and Liangcai Gao

36th Conference on Neural Information Processing Systems (NeurIPS 2022).

**Algorithm 1** Sequential algorithm

1: **Input:** graph $G = (V, E)$, filter function $f$.
2: Initialise-Nodes($V, f$)
3: $Q = $ Sort-Queue($V$)
4: **while** Q is not empty **do**
5:  $u = Q$.pop-min()
6:  **for** $v \in G$.neighbors($u$) **do**
7:   Relax-Edge($u, v, f$)
8:  **end for**
9: **end while**

---

**Algorithm 2** Computation of EPD

1: **Input:** filter function $f$, input graph $G = (V, E)$
2: $V, E = $ sorted($V, E, f$)
3: $PD_0 = $ Union-Find($V, E, f$), $PD_1 = \{\}$
4: **for** $i \in V$ **do**
5:  $C_i = \{C_{i_j} | (i, j) \in E, f(j) > f(i)\}$, $E_i = E$
6:  **for** $C_{i_j} \in C_i$ **do**
7:   $f(C_{i_j}) = f(i)$, $E_i = E_i - \{(i, j)\} + \{(C_{i_j}, j)\}$
8:  **end for**
9:  $PD_1^i = $ Union-Find-step($V + C_i - \{i\}, E_i, f, C_i$)
10:  $PD_1 += PD_1^i$
11: **end for**
12: **Output:** $PD_0, PD_1$

---

**Algorithm 3** Union-Find-step (Sequential)

1: **Input:** $V, E, f, C_i$
2: $PD_1^i = \{\}$
3: **for** $v \in V$ **do**
4:  $v.value = f(v)$, $v.root = v$
5: **end for**
6: $Q = $ Sort($V$), $Q = Q - \{v | f(v) < f(i)\}$, $G = \{Q, E_Q\}$, where $E_Q = E \cup Q^2$.
7: **while** Q is not empty **do**
8:  $u = Q$.pop-min()
9:  **for** $v \in G$.neighbors($u$) **do**
10:
11:   $pu, pv = $ Find-Root($u$), Find-Root($v$)
12:   **if** $pu \neq pv$ **then**
13:    $s = argmin(pu.value, pv.value)$
14:    $l = argmax(pu.value, pv.value)$
15:    $l.root = s$
16:    **if** $pu \in C_i$ and $pv \in C_i$ **then**
17:     $PD_1^i + \{(u.value, l.value)\}$
18:    **end if**
19:   **end if**
20:  **end for**
21: **end while**
22: **Function:** Find-Root($u$)
23: $pu = u$
24: **while** $pu \neq pu.root$ **do**
25:  $pu.root = (pu.root).root$, $pu = pu.root$
26: **end while**
27: **Return:** $pu$

---

time [2, 16]. Nevertheless, this algorithm remains theoretical, and in practice, often a quadratic $O(|V||E|)$ time algorithm is used for its simplicity [44]. Recently, some works have been proposed to accelerate the computation in a data-driven manner [36, 27, 50, 11]. However, these works try to estimate the persistence image [1], a coarsened topological feature rather than the persistence diagram itself, leading to much worse performance in both approximation error and downstream tasks. Compared with previous works, we propose a novel framework that directly predicts extended persistence diagrams on graphs. As shown in the experiment, the proposed model has achieved a satisfying approximation error while remaining a high efficiency as well.

## 0.2 Complexity and Correctness of Algorithm 2

In this section, we show the complexity and the correctness of Algorithm 2.

### 0.2.1 Complexity

The computational complexity of the Union-Find algorithm is $O(|E|\alpha(|E|))$ [9], where $\alpha(\cdot)$ is the inverse Ackermann function. Therefore, we need $O(|V||E|\alpha(|E|))$ time to compute an 1D EPD using Algorithm 2. Note this sequential algorithm is not necessarily the most efficient one. In practice, one may use the quadratic algorithm ($O(|V||E|)$) as in [44]. We also note that although not formally published, the best known algorithm for EPD computation is quasilinear, $O(|E| \log |V|)$, using the data structure of mergeable trees [2, 16]. But this algorithm remains theoretical so far.

### 0.2.2 Correctness

Formally, we restate the theorem below (The theorem is named Theorem 3.1 in the main paper). For a clear statement, we present the standard EPD computation algorithm in Algorithm 4. The

detailed description of Algorithm 4 is beyond the scope of the paper. We only introduce the needed information, and refer the readers to [8, 14] for details.

**Theorem 0.1.** *Algorithm 2 outputs the same 1D EPDs as Algorithm 4.*

As stated in Section 2 and Section 3 in the paper, for an edge (1-simplex) $e \in E$, it is either paired with a vertex or an edge. In the former case, the edge, defined as a negative edge, kills a connected component, and gives rise to a 0D persistence point. In the latter case, the edge, defined as a positive edge (in the ascending filtration), creates a loop during the ascending filtration. The loop will ultimately be killed by another edge during the descending filtration (defined as a positive edge in the descending filtration). Hence the positive edge in the ascending filtration is paired with a positive edge in the descending filtration, and gives rise to a 1D extended persistence point. For simplicity, we will call the positive edges in the ascending filtration as *ascending positive edges*, and the positive edges in the descending filtration as *descending positive edges*.

In other words, to compute the 1D EPDs, we can simply find the pairing partner for all positive edges. In the following paragraphs, we show that Algorithm 2 produces the same extended persistence pair as the standard EPD computation algorithm. We first present a definition of the "thinnest pair":

**Thinnest pair.** Given a filter function $f : X \rightarrow \mathcal{R}$, the pair of edges $(e_1, e_2)$ with $f(e_1) < f(e_2)$ is defined as the thinnest pair if the following condition is satisfied: (1) there is a cycle $C$ having $e_1$ as the lowest edge, and $e_2$ as the highest edge; (2) for any other cycle with $e_1$ as the lowest edge, if its highest edge $e_2$ satisfies that $f(e_3) \neq f(e_2)$, then $f(e_3) > f(e_2)$. Symmetrically, among all cycles having $e_2$ as the highest edge, $e_1$ is the lowest edge in a cycle such that this lowest value is the highest possible.

**Lemma 0.2.** *For every ascending positive edge, Algorithm 2 finds its "thinnest pair".*

*Proof.* Algorithm 2 decomposes the 1D extended persistence pair finding for all edges into pair-finding among all nodes. In particular, for a given node $u$, it uses Algorithm 3 to find the pair for its upper edges. There are two cases:

**Case 1.** If the upper edge is an ascending negative edge, then it will kill a connected component, and will not influence the 1D extended persistence pairing.

**Case 2.** If the upper edge is an ascending positive edge, it will be paired with the loop once the loop is created in the union-find process. The edge, called $e$, is the lowest edge in the loop, called $C$. Recall that $C$ is also the first loop that appears in the union-find process with $e$ as the lowest edge. Therefore $C$ is guaranteed to contain the highest value which is the lowest possible. According to the definition, this will lead to the "thinnest pair"[2]. $\square$

---

**Algorithm 4** The standard EPD computation algorithm

---

1: **Input:** filter funtion $f$, input graph $G$
2: $EPD = \{\}$
3: $M = $ build reduction matrix$(f, G)$, where $M$ is a $2m * 2m$ binary matrix.
4: **for** $j = 1$ **to** $2m$ **do**
5:    **while** $\exists k < j$ with $low_M(k) = low_M(j)$ **do**
6:       add column $k$ to column $j$
7:    **end while**
8:    add $(f(low_M(j)), f(j))$ to $EPD$
9: **end for**
10: **Output:** $EPD$

---

**Lemma 0.3.** *In the descending filtration of Algorithm 4, an edge $e$ is paired if a loop $C$ has already appeared, with $e$ as its lowest edge.*

*Proof.* Every column/row of the binary matrix $M$ shown in Algorithm 4 corresponds to a simplex (node/edge) in the input graph $G$. For simplicity, we replace the index in $M$ with the simplex it represents in the rest of the paper. For an edge $e$, $low_M(e)$ denotes its lowest row $e_1$, with $M[e, e_1] = 1$. After the matrix reduction process, $e$ and $e_1$ will form an extended persistence pair.

---

[2]We note that once a loop appears in the union-find process, it will consist of two different upper edges. Considering that the two upper edges share the same filter value in the ascending filtration, the output persistence point will not change no matter which edge is paired with the loop.

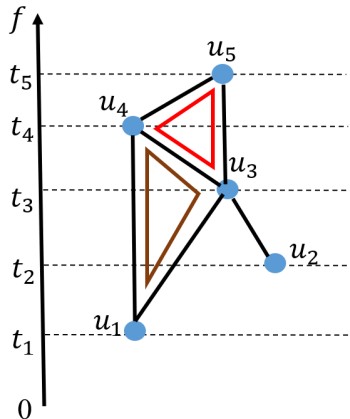

Figure 1: A toy example for Lemma 0.4.

It has been shown in [8, 44] that for a loop, its highest edge and lowest edge form its extended persistence pair. In other words, $e$ and $e_1$ are the lowest and highest edges of the loop they form. Assume that a loop $C$ has already aroused with $e$ as its lowest edge, then there are two cases for the highest edge $e_1$ in $C$:

**Case 1.** If there does not exist an edge $e_2$, that appears before $e_1$ in the descending filtration, with $low_M(e_2) = e = low_M(e_1)$, then $e$ will be paired with $e_1$ in Algorithm 4.

**Case 2.** If there exists an edge $e_2$, that appears before $e_1$ in the descending filtration, with $low_M(e_2) = e = low_M(e_1)$, then $e$ will be paired with $e_2$ or even other edges that appears earlier than $e_2$. Among all possibilities, $e$ is paired before $C$ appears in Algorithm 4.

In other words, $e$ will be paired with $e_1$ or before $e_1$ in Algorithm 4.

□

**Lemma 0.4.** *Algorithm 4 finds the "thinnest pair" for all positive edges.*

*Proof.* During the descending filtration, when an edge $e = u_i u_j$ appears, there are two cases:

**Case 1.** $e$ is a negative edge, and will kill a connected component. This will not influence the 1D extended persistence pair.

**Case 2.** $e$ is a positive edge, and will be paired with an ascending positive edge $e_1 = u_a u_b$. Assume that $e_1$ does not construct the "thinnest pair" with $e$, then there exists an edge $e_2 = u_c u_d$, that forms the "thinnest pair" with $e$. We can observe a loop $C = u_a \rightarrow u_c u_d \rightarrow u_b$, in which all edges born earlier than $e$ in the descending filtration, and $e_1$ is the latest edge in the ascending filtration. A toy example is shown in Figure 1, where $e = u_1 u_3$, $e_1 = u_4 u_5$, $e_2 = u_3 u_4$, and $C$ is the red loop. Then according to Lemma 0.3, $e_1$ will be paired no later than $C$ appears. In other words, it has already been paired before $e$ appears. Therefore, the assumption is wrong, and Algorithm 4 will find the "thinnest pair" for all positive edges.

□

According to Lemma 0.2 and Lemma 0.4, Algorithm 2 will produce the same 1D extended persistence pair as Algorithm 4. Therefore, they output the same 1D EPD.

## 0.3 Union-Find Algorithm

In this section, we rewrite the well-known Union-Find algorithm [9] in a sequential format. The algorithm is listed in Algorithm 5. Therefore we can use the proposed framework to estimate $PD_0$.

Table 1: Statistics of the node classification datasets

| Dataset | Classes | Nodes | Edges | Features | Avg degree |
|---------|---------|-------|-------|----------|------------|
| Cora | 7 | 2708 | 5429 | 1433 | 2.00 |
| Citeseer | 6 | 3327 | 4732 | 3703 | 1.42 |
| PubMed | 3 | 19717 | 44338 | 500 | 2.25 |
| CS | 15 | 18333 | 100227 | 6805 | 5.47 |
| Physics | 5 | 34493 | 282455 | 8415 | 8.19 |
| Computers | 10 | 13381 | 259159 | 767 | 19.37 |
| Photo | 8 | 7487 | 126530 | 745 | 16.90 |

---

**Algorithm 5** Union-Find (Sequential)

---

1: **Input:** $G = (V, E)$, $f$
2: $PD_0 = \{\}$
3: **for** $v \in V$ **do**
4:   $v.value = f(v)$, $v.root = v$
5: **end for**
6: $Q = \text{Sort}(V)$
7: **while** Q is not empty **do**
8:   $u = Q.\text{pop-min}()$
9:   **for** $v \in G.\text{neighbors}(u)$ **do**
10:    $pu, pv = \text{Find-Root}(u), \text{Find-Root}(v)$
11:    **if** $pu \neq pv$ **then**
12:     $s/l = argmin/argmax(pu.value, pv.value)$
13:     $l.root = s$
14:     $PD_0 + \{(l.value, u.value)\}$
15:    **end if**
16:   **end for**
17: **end while**
18: **Function:** $Find - Root(u)$
19: $pu = u$
20: **while** $pu \neq pu.root$ **do**
21:   $pu.root = (pu.root).root$, $pu = pu.root$
22: **end while**
23: **Return:** $pu$

---

## 0.4 Experimental details

### 0.4.1 Datasets.

In this paper, we exploit real-world datasets including:

1. Citation networks: Cora, Citeseer, and PubMed [34] are standard citation networks where nodes denote scientific documents and edges denote citation links.

2. Amazon shopping records: In Photo and Computers [35], nodes represent goods, edges represent that two goods are frequently brought together, and the node features are bag-of-words vectors.

3. Coauthor datasets: In CS and Physics [35], nodes denote authors and edges denote that the two authors co-author a paper.

The detailed statistics are available in Table 1.

### 0.4.2 Experimental Details

In this section, we mainly present the experimental settings on neural estimation, as for the setting in downstream graph representation learning tasks, we are consistent with [48, 44].

Following the settings in [48, 44], we extract 2-hop neighborhoods of all the nodes in Cora, Citeseer, PubMed and 1-hop neighborhoods of all the nodes in Photo, Computers, Physics, and CS. In the training process, we only adopt the $W_2$ distance between the predicted diagram and the ground truth diagram as the loss function, while the PIE between the predicted persistence image and the ground truth persistence image only serves as an evaluation metric.

We adopt Adam as the optimizer with the learning rate set to 0.002 and weight decay set to 0.01. We build a 4-layer GNN framework with dropout set to 0. In the training process, we set the batch size to 10, and the training epoch to 20. In this paper, we also exploit a 2-layer MLP to transform the node embedding obtained by the GNN to the persistence points on edges. In the framework, PRELU is

Table 2: Transferability in terms of different graph structures ($W_2$ distance.)

| Pre-train | Cora | Citeseer | PubMed | Photo | Computers |
|-----------|-------|----------|--------|-------|-----------|
| Pre-train | 0.392 | 0.279 | 0.444 | 0.379 | 0.404 |
| Fine-tune | 0.348 | 0.259 | 0.360 | 0.380 | 0.381 |
| Standard | 0.354 | 0.267 | 0.344 | 0.379 | 0.377 |

adopted as the activation function, the dimension of hidden layers is set to 32, and the dimension of the output persistence image is 25. All the experiments are implemented with two Intel Xeon Gold 5128 processors,192GB RAM, and 10 NVIDIA 2080TI graphics cards.

Notice that in the normal computation of Wasserstein distance between PDs, the persistence points can be paired to the diagonal or the persistence points in the other diagram. However, in the experiments, we observe that with this loss function as the supervision, the model may converge to local minima, e.g., all the predicted persistence points are paired to diagonal. Therefore, the predicted points all converge to the diagonal and contain no topological information. To avoid such situations, we force the predicted points to pair with the persistence points in the ground truth diagram rather than the diagonal in the training stage. In the reference stage, we report the normal $W_2$ distance between persistence diagrams, that is, to let the predicted points pair with the diagonal.

### 0.4.3 About the assets we used

Our model is experimented on benchmarks from [28, 21, 13, 34, 35] provided under MIT license.

## 0.5 Additional Experiments

### 0.5.1 Experiments on transferability

In this section, we design experiments to evaluate the transferability of PDGNN in terms of different graph structures. Our aim is to evaluate whether the pre-trained model can estimate EPDs on totally unseen graphs. Therefore, we evaluate the models pre-trained on Photo on other datasets, and report the $W_2$ distance between the predicted diagrams and ground truth EPDs. Notice that we only use Ollivier-Ricci curvature [30] as the filter function. The results are shown in Table 2.

In Table 2, "Pre-train" is to directly predict the EPDs with the pre-trained model, and "Fine-tune" is to fine-tune an epoch on the new datasets, and then predict the EPDs. As shown in Table 2, directly predicting the EPDs with the pre-trained model perform comparably with the standard settings among datasets. We also observe that with only a one-epoch fine-tuning, the pre-trained model can achieve almost an equal performance compared with the standard setting. It justifies the fine transferability of PDGNN. Therefore, in a totally new environment, instead of training the uninitialized models for many epochs, we can simply fine-tune or even directly use the pre-trained model to estimate EPDs on new graph structures.

### 0.5.2 Evaluation on the influence of training samples

In this section, we evaluate the influence of training samples on PDGNN. We aim to show that the model can reach an acceptable performance with only a small number of training samples.

Recall that for a given graph, we extract the $k$-hop neighborhoods of all the nodes and randomly select 80% of these vicinity graphs to train PDGNN. For a thorough evaluation, we train PDGNN with 5/10/20/40% vicinity graphs in this experiment and report the $W_2$ distance of persistence diagrams, the PIE of persistence images, and the node classification accuracy (NCA) in Table 3. We also visualize the influence in Figure 2 and Figure 3.

As shown in Figure 2, the training error tends to converge as the training samples gradually increase. Considering that the $W_2$ distance and PIE cannot directly reflect the learning power as NCA does, we select a vicinity graph in Cora which is hard for PDGNN to learn and visualize in Figure 3. As shown in the figure, as the number of training samples increases, we find that PDGNN can gradually capture the ground truth persistence points in the up y-axis and the up-right diagonal with much less noise. The number of training samples may help the model learn the hard samples better.

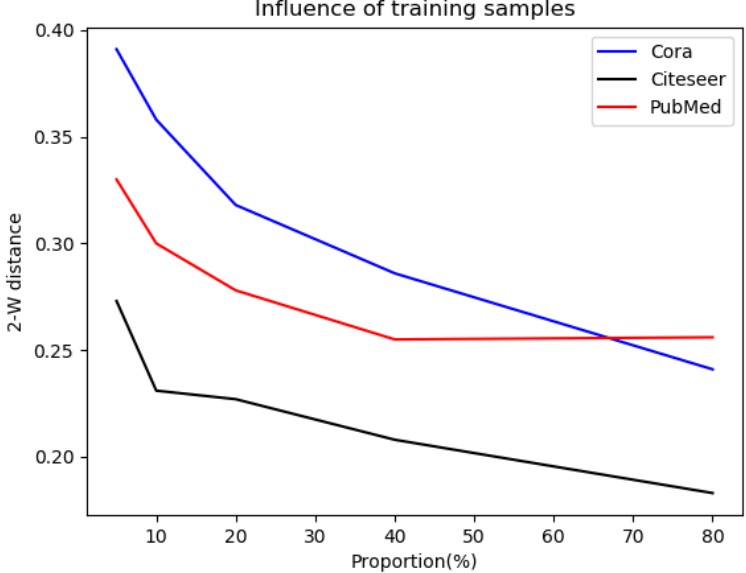

Figure 2: Influence of training samples.

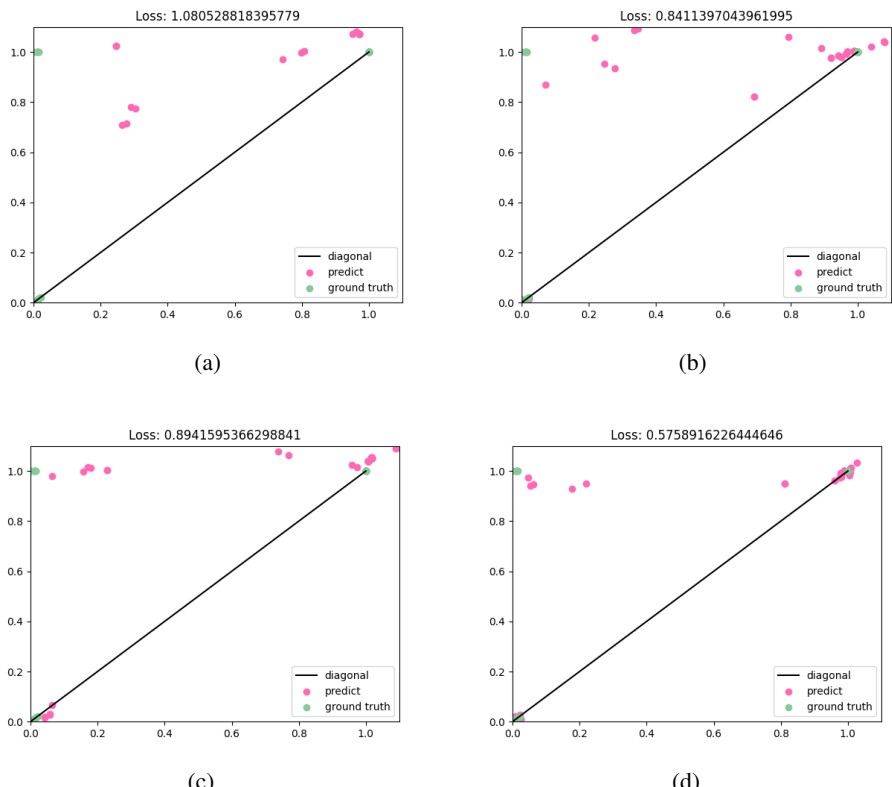

Figure 3: Visualization on the influence of training samples. We select a vicinity graph in Cora with Ollivier-Ricci curvature as the filter function, and plot the influence of training samples on the $W_2$ distance (loss) of EPDs. (a), (b), and (c) denote the prediction of PDGNN with 5/10/20% training samples, (d) denotes the prediction of PDGNN with the standard setting.

Table 3: Influence of training samples on PDGNN

| Dataset | Cora | | | Citeseer | | | PubMed | | |
|---|---|---|---|---|---|---|---|---|---|
| Proportion | $W_2$ | PIE | NCA | $W_2$ | PIE | NCA | $W_2$ | PIE | NCA |
| 5% | 0.391 | 2.51e-3 | 81.3±0.6 | 0.273 | 3.12e-3 | 70.0±0.7 | 0.330 | 4.35e-3 | 78.0±0.4 |
| 10% | 0.358 | 1.88e-3 | 81.6±0.7 | 0.231 | 3.01e-3 | 70.5±0.5 | 0.300 | 2.36e-3 | 78.5±0.4 |
| 20% | 0.318 | 6.99e-4 | 81.8±0.8 | 0.227 | 1.63e-3 | 70.6±0.5 | 0.278 | 1.03e-3 | 78.3±0.3 |
| 40% | 0.286 | 9.79e-4 | 81.6±0.6 | 0.208 | 9.98e-4 | 70.9±0.6 | 0.255 | 1.34e-3 | 78.8±0.5 |
| 80% | 0.241 | 4.75e-4 | 82.0±0.5 | 0.183 | 4.43e-4 | 70.8±0.5 | 0.256 | 8.95e-4 | 78.7±0.6 |

Table 4: Statistics and approximation error on the graph classification datasets

| Dataset | Graphs | Avg Nodes | Avg Edges | $W_2$ | PIE |
|---|---|---|---|---|---|
| MUTAG | 188 | 17.9 | 39.6 | 0.300 | 3.06e-4 |
| ENZYMES | 600 | 32.6 | 124.3 | 0.299 | 3.72e-3 |
| PROTEINS | 1113 | 39.1 | 145.6 | 0.194 | 8.30e-4 |
| COLLAB | 5000 | 74.5 | 4914.4 | 0.346 | 3.25e-2 |
| IMDB-BINARY | 1000 | 19.8 | 193.1 | 0.176 | 4.13e-4 |
| REDDIT-BINARY | 2000 | 429.6 | 995.5 | 0.383 | 1.92e-4 |
| ZINC (subset) | 12000 | 23.2 | 49.8 | 0.089 | 1.52e-5 |
| OGBG-MolHIV | 41127 | 25.5 | 27.5 | 0.104 | 4.96e-5 |

We also observe that in Table 3, PDGNN reaches a comparable performance on NCA with much fewer training samples. The observation shows that a little perturbation on the persistence image will not influence its structural information very much.

Combining the observation in Section 0.5.1 and Section 0.5.2, we can safely conclude that our model can be easily generalized to other frameworks. PDGNN does not need many training samples to reach an acceptable performance, and it can be easily transferred to totally unseen graphs.

### 0.5.3   Experiments on graph classification datasets.

In the experiment part, we only consider predicting EPDs of the $k$-hop neighborhoods of the original graphs. Even if these vicinity graphs can be large and dense, there can be structural differences between these vicinity graphs and other real-world graphs. In this section, we do further experiments on graph classification datasets, in which we approximate the EPDs of the real-world graphs rather than the vicinity graphs. We exploit various datasets from the TU Dortmund University [28], benchmarking-GNN [13], and OGB [21]. The detailed information of these datasets and the approximation error are all available in Table 4.

Notice that we do not add Ollivier-Ricci curvature as the filter function here, because computing the filter function on all the graphs will bring too much computational cost. Comparing the results from Table 4 and the results on vicinity graphs, we observe that the performance on graph classification datasets is slightly worse than the performance on vicinity graphs. This may be due to the fact that in graph classification datasets, the training samples can be very small, e.g., there are only 188 graphs in MUTAG, therefore the training is under-fit. On the contrary, the satisfying approximation quality on OGBG-MolHIV and ZINC can be due to their large number of training samples.

To evaluate the results more clearly, we also visualize some selected examples in Figure 4. As shown in the figure, in most situations, PDGNN can well estimate the EPDs on these graphs, and the $W_2$ distance around 0.3 is generally an acceptable result.

### 0.5.4   Why not directly approximating PIs

We believe directly estimated PIs will lose important structural information that can be crucial for downstream tasks. PI is only an approximation of the persistence diagram. The L2 distance between PIs does not accurately reflect the true Wasserstein distance between diagrams. Therefore, using an L2-distance-based loss to directly learn the PI may lead to the loss of important structural information carried by a diagram. An example is provided in Figure 5. For a sample vicinity graph from Cora, we compare the ground truth PI (computed from the ground truth diagram), the PI computed from the

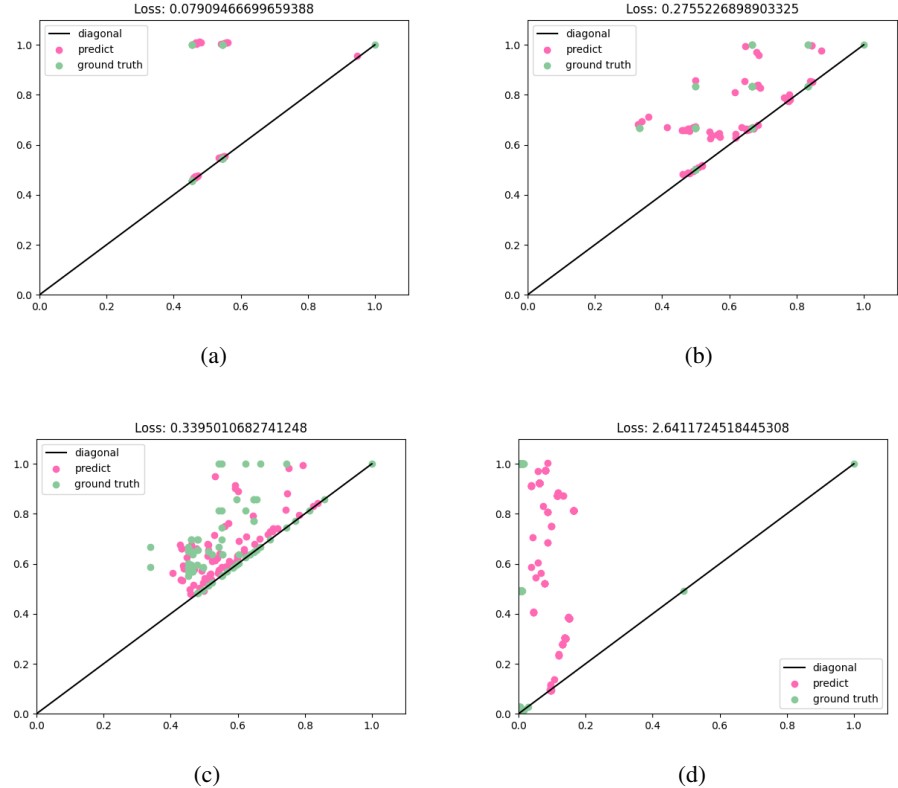

(a)

(b)

(c)

(d)

Figure 4: Visualization of graph classification samples. We select samples from IMDB-BINARY, PROTEINS, ENZYMES, and REDDIT-BINARY, respectively, and report the $W_2$ distance (loss).

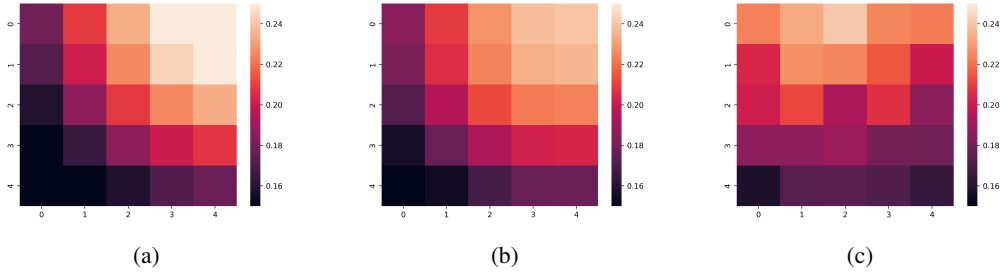

(a)

(b)

(c)

Figure 5: Examples to explain why not directly approximating PIs.

diagram estimated by our method PDGNN, and the PI directly estimated by GIN. Both estimated PIs have similar L2 distances from the ground truth PI. But we observe that the PI estimated by PDGNN has a very similar spatial distribution to the ground truth PI. This structural property, however, is not preserved by the directly estimated PI. Such loss of structural information of directly estimated PIs, although not captured by the L2 error, partially explains their worse representation power. In Table 2 and Table 3 in the main paper, the directly estimated PIs (PEGN(GIN_PI) and TLC-GNN(GIN_PI)) perform worse in the downstream task.

This choice of estimating diagrams instead of PIs is a part of the overarching theme of our paper. Note that the main contribution of our paper is to transfer a complicated and uncontrollable learning process to a controllable process with algorithmic insight. This general principle also applies to our learning algorithm. We decompose the diagram computation algorithm into sub-algorithms, which can be approximated well by a GNN. This is the reason PDGNN approximates the diagrams much better than other baselines in Table 1 in the main paper.

Table 5: Experiments on large and sparse datasets.

| Dataset | Cora | Citeseer | PubMed |
|---|---|---|---|
| Node | 2485 | 2120 | 19717 |
| Edge | 5069 | 3679 | 44324 |
| Fast [44] | 0.184 | 0.068 | 1.816 |
| Gudhi [37] | 0.045 | 0.023 | 1.696 |
| PDGNN | **0.006** | **0.005** | **0.007** |

Table 6: Experiments on the choice of filter functions.

| Dataset | Cora | | Citeseer | | PubMed | |
|---|---|---|---|---|---|---|
| Filter | clustering | centrality | clustering | centrality | clustering | centrality |
| Evaluation on approximation error | | | | | | |
| $W_2$ | 0.392 | 0.332 | 0.178 | 0.237 | 0.267 | 0.322 |
| PIE | 1.53e-3 | 4.14e-4 | 7.17e-4 | 4.65e-4 | 2.5e-3 | 3.75e-4 |
| Evaluation on Time (s) | | | | | | |
| Fast [44] | 2.10 | 2.21 | 1.16 | 1.31 | 38.07 | 39.05 |
| Gudhi [37] | 0.98 | 1.00 | 0.59 | 0.63 | 16.79 | 16.24 |
| PDGNN | 11.25 | 11.30 | 13.61 | 13.62 | 66.19 | 67.29 |

### 0.5.5   Experiments on large and sparse datasets.

In the experiments in the main paper, the input is the $k$-hop vicinity graphs. On citation graphs, the vicinity graph remains small. On these small graphs, the exact sequential algorithm like Gudhi has less overhead, and thus is unsurprisingly faster.

Indeed, on large and sparse graphs, our method outperforms strong baselines like Gudhi significantly. In Table 5, we compare the running time (in seconds) on popular citation networks including Cora, Citeceer, and PubMed. For each graph, we run experiments on the largest connected subgraph. We also report the number of nodes/edges of the selected subgraph.

### 0.5.6   Experiment on the choice of filter functions/other graph metrics.

In Table 6, we set degree centrality and clustering coefficient as the filter function, follow the settings in Table 1 and report the approximation error on Cora, Citeseer, and PubMed. We also report computation time following the setting in Table 4. The only difference is that below we report the time to generate all vicinity graphs (rather than 1000 graphs as in Table 4).

We observe that (1) the filter function only has a minor influence on inference/computation speed, for both the sequential algorithm and ours; (2) the filter function does influence the approximation error. The reason is that different filter functions have different ranges; functions with larger ranges tend to have larger approximation errors, especially on PIE. This is another evidence that the distance function on PIs is not very robust for learning.

### 0.5.7   Experiments on the threshold value of average node/edge to decide which method is the fastest to compute/estimate EPDs.

To find the threshold, we use the well-known Stochastic Block Model (SBM) [20] to generate synthetic graphs. We set the number of nodes in these synthetic graphs from 200 to 300, with 10 as the step. In these graphs, we randomly generate 5 different clusters, and set the probability of edges intra-cluster to 0.4, and the probability of edges inter-cluster to 0.1. In this way, we can obtain 11 graphs with different nodes and edges. We set node degree as the filter function, and add experiments on the largest connected components of these 11 graphs. The information of the selected connected

Table 7: Experiments on the threshold value.

| Node | 80 | 84 | 88 | 92 | 96 | 100 | 104 | 108 | 112 | 116 | 120 |
|---|---|---|---|---|---|---|---|---|---|---|---|
| Edge | 515 | 585 | 660 | 713 | 759 | 820 | 943 | 1012 | 1060 | 1152 | 1231 |
| Fast [44] | 6.8e-3 | 8.0e-3 | 8.9e-3 | 9.6e-3 | 1.0e-2 | 1.1e-2 | 1.3e-2 | 1.4e-2 | 1.4e-2 | 1.6e-2 | 1.7e-2 |
| Gudhi [37] | **2.5e-3** | **3.2e-3** | **3.6e-3** | **3.9e-3** | **4.0e-3** | 4.8e-3 | 5.5e-3 | 6.0e-3 | 6.4e-3 | 6.6e-3 | 6.8e-3 |
| PDGNN | 4.5e-3 | 4.5e-3 | 4.6e-3 | 4.6e-3 | 4.6e-3 | **4.6e-3** | **4.7e-3** | **4.7e-3** | **4.7e-3** | **4.7e-3** | **4.8e-3** |

Table 8: Experiments on the threshold value.

| Node | 100 | 100 | 100 | 100 | 100 | 100 | 100 | 100 | 100 | 100 | 100 |
|---|---|---|---|---|---|---|---|---|---|---|---|
| Edge | 489 | 529 | 595 | 652 | 766 | 842 | 968 | 1011 | 1082 | 1231 | 1307 |
| Fast [44] | 7.0e-3 | 7.4e-3 | 8.3e-3 | 9.0e-3 | 1.1e-2 | 1.2e-2 | 1.3e-2 | 1.4e-2 | 1.4e-2 | 1.6e-2 | 1.7e-2 |
| Gudhi [37] | **2.8e-3** | **2.9e-3** | **3.0e-3** | **4.1e-3** | **4.2e-3** | 5.1e-3 | 5.5e-3 | 5.9e-3 | 6.2e-3 | 6.3e-3 | 6.7e-3 |
| PDGNN | 4.1e-3 | 4.1e-3 | 4.2e-3 | 4.2e-3 | 4.3e-3 | **4.4e-3** | **4.7e-3** | **4.7e-3** | **4.8e-3** | **4.8e-3** | **4.8e-3** |

graphs and the running time (second) are listed in Table 7. As shown in the Table, the threshold is around 100 nodes / 820 edges.

We also evaluate the influence of density. We fix the node number of the SBM model to 250, and set the probability of edges intra-cluster from 0.5 to 0.7, and the probability of edges inter-cluster from 0.05 to 0.15. The steps for intra-cluster and inter-cluster are 0.02 and 0.01, respectively. In this way, we can obtain 11 graphs with the same nodes and different edges. We set node degree as the filter function, and add experiments on the largest connected components of these 11 graphs. The information of the selected connected graphs and the running time (second) are also listed in Table 8. As shown in the Table, the threshold is around 100 nodes / 766 edges.

### 0.5.8 Limitation of the paper.

First, in certain cases like Figure 4 (d), the model only captures a tendency of the EPD. This can be because that the distribution of the EPD of the selected graph is seldom in the training samples. Therefore, it is hard for the model to estimate these EPDs correctly.

Second, topological features are just one side of the data. In many cases, only using topological features such as EPDs to represent the information of graphs is not enough. A better way is to introduce other information such as the semantic information of graphs as complementary.