# OpenReview forum: "Neural Approximation of Graph Topological Features"
_NeurIPS.cc/2022/Conference — NeurIPS 2022 Accept_

### Official Review · Reviewer_8ijg · 2022-07-07

**Rating:** 7
**Confidence:** 4
**Soundness:** 3 good
**Presentation:** 4 excellent
**Contribution:** 3 good

**Summary:**

The extraction of topological features (called persistence diagrams - PDs) from structured data (such as graphs, point clouds, time series, etc.) is a promising approach to develop new machine learning methods but suffers from high computational complexity (among other things) to be faithfully used in learning pipelines. This happens mostly because the computation of PDs relies on combinatorial algorithms (roughly, sparse matrix reduction) that do not scale well with the size of the considered data.

Following a recent line of development, this article proposes to bypass the exact computation of topological features via combinatorial algorithms and to instead approximate it using neural networks. The difference with previous approaches is that instead of brutaly computing PDs using an ad-hoc architecture, authors decompose the (extended) PD computation in different algorithmic steps, which roughly rely on running a Union-Find. It invite them to design a Graph Neural Network (GNN) capable of learning to reproduce this sub-routine that they call PDGNN, essentially based on an adapted aggregation function (mixing a minimum and a maximum).

They showcase the benefits of the proposed approach on different experimental tasks.

**Questions:**

# Questions

1. In line 271, it is said that the training loss was the $W_2$ metric rather than the L2 norm between some vectorization such as persistence images. Given that the $W_2$ requires to compute a matching to retrieve a gradient, doesn't it involve some computational burden in practice?
2. (related to 1.) In line 49, it is said that "The Wasserstein distance can be naturally decomposed into supervision loss for each edge (...) improve learning efficiency" ; while I guess that I understand the underlying idea in an informal way, could you elaborate on this? In concrete terms, how is the training loss implemented and how does it benefits from the "edge decomposition" properties of your approach?
3. Given that Persistence Images should be stable wrt the $W_2$ metric, isn't it somewhat "redundant" to use the PIE as a second evaluation metric?

# Suggestions

- The works _Learning persistent homology of 3D point clouds_ and _RipsNet: a general architecture for fast and robust estimation of the persistent homology of point clouds_ are also learning to produce topological features (on top of point cloud this times) and may be referenced along PI-Net and _Can Neural Networks learn persistent homology features_ in the introduction (line 38).

**Limitations:**

I think the limitations of the work have been properly addressed (for instance the authors aknowledge that their approach is only beneficial on large graphs --- exact computation may be more efficient otherwise).

I do not identify potential negative societal impact _specific to this work_.

**Strengths And Weaknesses:**

- [Originality] While not being entirely novel for either the TDA nor the ML communities, the approach proposed in this paper is quite fresh and promising. At least from the TDA perspective, I think that the idea of "learning to produce topological descriptors **by approximating the inner-algorithms** in a relevant way" was somewhat in the minds, but this paper is the first one to do so in a convincing manner to the best of my knowledge.

- [Clarity] The paper is well written an pleasant to read.

-  [Significance] The approach developed in this work seems quite promising and may be a source of inspiration for future works in TDA, in particular showcasing the potential of GNN when it comes to extract topological information from data.

- [Quality] I think this paper is of great quality overall. Aside from the interesting approach, I also like the experimental evaluation methodology which covers the important questions (i) Is our model capable of producing outputs similar to ground truth PDs (Sec. 5.1), (ii) are our approximated PDs still relevant as topological features in downstream tasks, (iii) what are the computational benefits/drawbacks of using this approach in terms of running time.

---

> ### Author Response · Authors · 2022-08-02
> **Response to Reviewer 8ijg**
>
> Thanks very much for the constructive feedback and for pointing out the main idea of the paper “learning to produce topological descriptors by approximating the inner-algorithms in a relevant way”.
>
> **(1) How is the training Wasserstein loss implemented and how does it benefit from the "edge decomposition" properties of your approach?**
>
> Thanks for pointing out the confusion. We will clarify this in the paper.
>
> The key is that we can establish a one-to-one correspondence between edges and persistence pairs in the EPD. During the ascending filtration, each edge is either the destroyer of a 0D persistent homology class (i.e., connecting two connected components), or the creator of a 1D persistent homology class (i.e., creating a loop that will be destroyed during the descending filtration). This way, we can assign to each edge of the graph a unique persistence pair as its “label”. The computation of the EPD is then reduced into an edge label prediction problem – predicting the persistence pair associated with each edge.
>
> During training time, our model predicts persistence pairs for all edges. These pairs are collected as the predicted EPD. The predicted EPD is then compared with ground truth EPD using Wasserstein distance; this is the training loss. It is true that the computation of the Wasserstein distance is nontrivial and is indeed a bottleneck. But please note that this only happens during training. At the inference stage, predicting edge labels and then the entire EPD does not involve the expensive Wasserstein distance computation. Also note that we observe good transferability (Section 4.1 in the appendix) of our model, meaning we can apply a pre-trained model to a new graph without training.
>
>
> **(2) Isn't it somewhat "redundant" to use the PIE as a second evaluation metric?**
>
> We have to use PIE to evaluate some baseline methods [30, 39]. These methods directly approximate PIs, and thus cannot be evaluated using W2 distance.
>
> **(3) Related works on estimating persistent homology of point clouds.**
>
> Thank you for suggesting the references. We will cite and discuss them in the paper.

---

> > ### Comment · Reviewer_8ijg · 2022-08-03
> > **Still about the Wasserstein distance**
> >
> > Thank you for your answer. Here I some following comments (mainly about the training loss) :
> >
> > > During training time, our model predicts persistence pairs for all edges. These pairs are collected as the predicted EPD. The predicted EPD is then compared with ground truth EPD using Wasserstein distance; this is the training loss.
> >
> > Just to be sure my understanding is correct: it means that at training time, you compute an optimal matching between the collected pairs (that correspond to two "filtration values") and the target EPD, and then derive a gradient from this.
> > Isn't it how differentiation wrt the Wasserstein distance is already done (for instance in the work of Carrière et al., ICML 2021)? In this work, from my understanding, authors identify points in the PDs with couple (b,d) of birth-death times in the input filtration, and differentiate through it.
> > Anyway, that's definitely not a major concern, I was just wondering if I was missing something that would make the differentiation particularly efficient in your setting.
> >
> > >It is true that the computation of the Wasserstein distance is nontrivial and is indeed a bottleneck. But please note that this only happens during training.
> >
> > Sure, I agree this is not a major issue (though it shouldn't be ignored either).

---

> > > ### Author Response · Authors · 2022-08-03
> > > **Discussion on the Wasserstein distance**
> > >
> > > Thank you very much for the quick response. We would like to clarify the point in our experimental details.
> > >
> > > **(4) “You compute an optimal matching between the collected pairs (that correspond to two "filtration values") and the target EPD, and then derive a gradient from this”**
> > >
> > > Yes, you are correct. Through the matching, the gradient is passed to each predicted persistence pair. Since we have established the one-to-one correspondence between pairs and edges, the gradient is then passed to the corresponding edge, and contributes to the representation learning.
> > >
> > >
> > > **(5) "I agree this is not a major issue (though it shouldn't be ignored either)."**
> > >
> > > Yes, we agree.

---

### Official Review · Reviewer_cEes · 2022-07-11

**Rating:** 5
**Confidence:** 3
**Soundness:** 3 good
**Presentation:** 2 fair
**Contribution:** 3 good

**Summary:**

In this paper, the authors propose a mechanism to reduce the computation time, which is a major bottleneck of the extended persistence diagram that has recently been proven effective for graph structure learning tasks. The authors analyze the calculation algorithm of EPD, describe the key Union-Find algorithm in seqential, and approximate it with GNN to simplify the calculation. In their experiments, the authors evaluated the effect of EPD on graph analysis, the impact of pride of place, and computation time, and showed that EPD can analyze graphs efficiently and accurately, especially for large graphs.

**Questions:**

Can you clarify the following two points?
- Provide evidence that EPD is more effective than other TDA methods.
- Discussion on whether there is any other research on EPD acceleration. Also, there are other speed-up methods for TDAs that are not EPD, such as [Cufar], etc. Can you mention why these methods cannot be used for EPD?

[Cufar] MATIJA ČUFAR AND ŽIGA VIRK, FAST COMPUTATION OF PERSISTENT HOMOLOGY REPRESENTATIVES WITH INVOLUTED PERSISTENT HOMOLOGY, https://arxiv.org/abs/2105.03629

Also, instead of approximating part of the algorithm with a GNN as an approximate computation of the diagram or PI, one could directly approximate the entire flow with a GNN or NN, but what is the reason for approximating only part of it? I know there is a trade-off between approximate performance and computation time, as well as the amount of data required, but is there anything you can mention? Please mention any that you can, although they may be related to issues to be considered in the future.

**Limitations:**

The authors have clearly defined the application, clearly described the performance, and adequately addressed the limitation of their work.

**Strengths And Weaknesses:**

Strength
- Algorithms to improve the efficiency of PED calculations were precisely constructed by analyzing the structure of PED calculations.
- The authors evaluate the effect of PED and the speed-up effect of this algorithm, as well as the accuracy of the speed-up.

Weakness
- Although the subject is to improve the efficiency of EPD calculations, there are insufficient references to other acceleration methods.
- This method is an EPD-only speed-up method, assuming that EPD has high performance, but there is no mention of whether EPD is superior to other graph-oriented TDA methods. If EPD is inferior to other TDA methods, the method is less valuable from an AI perspective, i.e., for graph analysis.

Overall, the effect of graph EPD on non-traditional TDA-based methods and the speedup of EPD are accurately described, and the contribution is significant. On the other hand, the evaluation is based on a limited set of assumptions that favor the proposed method and can be viewed as unfair.

---

> ### Author Response · Authors · 2022-08-02
> **Response to Reviewer cEes**
>
> **(1) EPD compared with other graph-oriented TDA methods.**
>
> Sorry we are not sure which other TDA methods you are referring to. The learning power of EPDs has been well established in recent years. Recent works [5, 51, 57] have shown that EPDs can achieve SOTA performance in node and link prediction tasks. The success motivates accelerating the computation of diagrams, either through approximation [30, 39] or through better algorithm design [51]. We address this fundamental task in this paper.
>
> **(2) The reason to approximate part of the EPD computation algorithm**
>
>
> Empirically, it is evident that direct approximation of the entire algorithm is very difficult. Please refer to the poor performance of baselines in Table 1 (GIN_PI, GAT_PI, GAT, GAT+MIN). The original EPD computation algorithm [11] involves a sequence of modulo-2 additions of boundary matrix columns that may involve edges and nodes far apart. Both the untypical modulo-2 arithmetic operation and the highly global computation are extremely hard to learn even for a deep neural network.
> Our idea of algorithmic decomposition is not simply a trade-off between approximation performance and computation. We use our algorithmic insights to circumvent the aforementioned challenges. Specifically, we decompose the algorithm into a set of parallel Union-Finds sub-routines. The benefit is threefold: (1) each Union-Find is only related to the persistence pairs originating from one node, and thus is a relatively local computation; (2) the Union-Find algorithm does not involve untypical modulo-2 arithmetic. Instead, it follows a certain sequential algorithm pattern that is known to be well aligned with GNN [44,49]; (3) these Union-Finds share common algorithm steps and potentially share similar information flows. Thus they can be approximated altogether with one single run of GNN. This is extremely efficient in practice. These benefits guarantee the superior performance of our algorithm, in both approximation error and running time.
>
> **(3) The evaluation is based on a limited set of assumptions that favor the proposed method and can be viewed as unfair.**
>
> Sorry this comment is too vague for us to respond to. The way we use persistence diagrams for graph learning tasks has been well established in recent years [5, 9, 17, 51, 56, 57]. Combining topological features with GNN clearly outperforms the traditional way of directly feeding topological features into a classifier [19, 26]. Under such a premise, we evaluate the approximation error, running time, and representation quality of our method. On a broad spectrum of popular graph benchmarks, our method outperforms SOTA methods.
>
> **(4) Other speed-up methods, e.g., [Cufar and Virk 2021].**
>
> Thank you for the reference. We will cite and discuss it. In fact, [Cufar and Virk 2021] does not accelerate the computation of persistence diagrams. It speeds up the computation of explicit cycle representatives for each persistent homology class. In terms of computing the diagram alone, it is not faster than known sequential matrix reduction algorithms. It is true that cycle representatives carry additional information and can potentially enrich persistence diagrams. We will be happy to explore how this can be leveraged for graph learning tasks in the future.

---

> ### Comment · Reviewer_cEes · 2022-08-03
> **Thanks for your response**
>
> First, I apologize for any misunderstanding I may have caused you. For formatting purposes, we listed our concerns in "Strengths And Weaknesses" and provided concrete suggestions for resolving those concerns in "Questions. After "overall", I write my impression of the reason for rating.
> Therefore, it is sufficient to answer the questions in the "Questions" section. In other words, it is enough to rebute the contents of "weakness".
>
> With regard to my second question, I think the responses (1) and (4), as well as the responses to the references pointed out by the other two reviewers (which I was not aware of), are sufficient.
> As to my additional questions, I understand your views. Although future research may yield different findings, I recognized that this is a reasonable policy at this time.
>
> As for the first question, I think I did not fully convey my intention. This method assumes that EPD using the filtration shown in Fig. 1 is excellent for graph analysis. However, it is difficult to understand why only the filtration shown in fig.1 is targeted. For example, PersLay [5] uses a heat kernel. If other filtration methods are better, the value of the proposed method will be smaller. So I requested evidence that the filtration in Fig. 1 is very effective. My description of this point was problematic, so if you are short on time, just speeding up the method using the proposed filtration is certainly worth enough, so I will evaluate it from that perspective. If possible, please provide evidence. I will re-evaluate after receiving the answer to this question.

---

> > ### Author Response · Authors · 2022-08-03
> > **Response to Reviewer cEes**
> >
> > **(5) It is difficult to understand why only the filtration shown in Fig.1 is targeted**
> >
> > Thank you very much for the quick response. We would like to clarify that the height-function-based filtration in Fig.1 is only for illustration purposes. We will add clarification in the caption of Fig.1.
> >
> > Our method applies to any filter function that can be assigned to a graph. In the paper, we cover 3 types of filter functions from state-of-the-art works, that is Ollivier-Ricci curvature [57, 51], heat kernel signature [5], and the node degree [17]. Details of these filter functions can be found in lines 255-256. When reporting approximation errors (Table 1), we report the mean score over diagrams of all filter functions. In addition, in our post-review response to Reviewer 5254 Q4, we add evaluations on 2 more filter functions: the clustering coefficient and the centrality. The empirical evidence shows that our method works well on all 5 types of filter functions.

---

> > > ### Comment · Reviewer_cEes · 2022-08-04
> > > **Thanks for the clarification**
> > >
> > > Thank you for the clarification. I found it a very useful tool. Although this issue seems to be in the field of TDA, it is also useful in the field of AI, so I think it is fully acceptable.

---

> > > > ### Author Response · Authors · 2022-08-04
> > > > **Thanks for your support**
> > > >
> > > > Thanks very much for your support!

---

### Official Review · Reviewer_5254 · 2022-07-12

**Rating:** 5
**Confidence:** 3
**Soundness:** 2 fair
**Presentation:** 3 good
**Contribution:** 2 fair

**Summary:**

In this paper, the authors propose an efficient learning framework to estimate extended persistence diagrams (EPD). It decomposes a reformulated version of EPD as an edge-wise prediction task into independent pairing problems. The authors use GNN architecture to learn the union-find algorithm which can solve the pairing problems.

**Questions:**

1. Can such EPD decomposition be extended to simplicial complexes (with 2-simplices, 3-simplices and more) apart from graph representations as simplicial complexes? If so, it would be interesting to also apply the proposed framework on some real-world higher-order data.
2. How is the value of k (k-hop) chosen when extracting the |V| vicinity graphs?
3.  The authors claim that their estimation method is much faster on large and dense datasets. Is there a threshold value of average N/E or degree to decide which method is the fastest to compute/estimate the EPD? Some additional experiment on real-world or synthetic graphs and a plot of computation time vs graph density may give an answer to this. The efficiency experiment is based on 2-hop neighborhoods. Does the value choice of k-hop affect such results?

**Limitations:**

See question 1.

**Strengths And Weaknesses:**

Pros:

1. The EPD approximation method is examined on varying datasets and achieves the best performance.
2. The proposed method is efficient on large and dense graphs.
3. The paper is well written and easy to follow.

Cons:

1. I suggest the authors check this related paper on using PD to learn graph representation and consider it as a baseline: https://openreview.net/forum?id=yqPnIRhHtZv
2. Large scale real-world graphs are usually sparse such as the brain connectome etc. However, in this paper the large graphs are mostly denser than the smaller ones. The authors claim that their estimation approach of EPD is faster than other methods on large and dense graphs, it would be worthwhile to check the performance on large sparse graphs, which are more common in real-world complex systems. The proposed method is actually slower when the graph is sparse (citation networks).
3. The major contribution of this paper is the efficient estimation of EPD using decomposition and GNN architecture to learn the reformulated edge prediction task. Downstream tasks in section 5.2 are more like an indirect way to examine the approximation accuracy. As long as the EPD approximation error is low, PEGN/TLC-GNN (True Diagram) and PEGN/TLC-GNN (PDGNN) will have similar performance. Therefore I recommend the authors to focus more on the EPD estimation experiments. For example, the author could further discuss and analyze why indirectly estimating the persistence image (PI) by estimating the EPD is better than directly estimating the PI to explain their experiment result presented in table 1. Another thing worthy to look at is the choice of filter functions. Ablation study or discovery of other graph metrics such as clustering coefficient, centrality can help to make the experiment complete.

---

> ### Author Response · Authors · 2022-08-02
> **Response to Reviewer 5254**
>
> **(1) Additional Downstream Tasks Baselines (“Permanifold” at https://github.com/pkyriakis/permanifold).**
>
> Thanks for your suggestion. This paper proposes a new feature representation of persistence diagrams for learning tasks. We are happy to discuss it and use it as a baseline downstream method to verify the learning power of our estimated diagrams. We use the original code (https://github.com/pkyriakis/permanifold) and try our best to reproduce the results (details are explained later). We compare the original method, Permanifold, with Permanifold-PDGNN, in which we replace the original diagrams with the estimation of our method PDGNN. The table below shows that the two methods achieve similar classification performance. This confirms that the representation power of our estimated diagrams is on par with the original diagrams.
>
>
> | |  IMDB-BINARY | IMDB-MULT |
> | :----:| :----: | :----: |
> |Permanifold | $74.20\pm1.60$ | $52.59\pm0.56$ |
> |Permanifold-PDGNN| $73.50\pm0.45$ |$ 51.06\pm0.74$|
>
> **Experimental details.** We use the original repository of Permanifold (https://github.com/pkyriakis/permanifold). Note that in the repository, diagrams are generated using the package cechmate (for filter functions such as the node degree). However, it requires a specific version of package phat=1.5.0a0, which is not available through pip or conda. Therefore, we change the PD generation package from cechmate to another well-known package gudhi [41].
> We follow the settings from the paper: to select the hyper-parameters from “manifold = [Euclidean, Poincare], manifold dimension = [3,6,9,10],  projection bases = [200, 350, 500], batch size = 64, epoch = 100” (other settings are the default setting of the code). We run the experiment with each set of hyper-parameters once and report the best one as the result. For IMDB-BINARY, we adopt the manifold as Euclidean, the manifold dimension as 3, and the projection bases as 500.  For IMDB-MULTI, we select the manifold as Euclidean, the manifold dimension as 9, and the projection bases as 500.
>
> **(2) Discuss and analyze why indirectly estimating the persistence image (PI) by estimating the EPD is better than directly estimating the PI**
>
> We believe directly estimated PIs will lose important structural information that can be crucial for downstream tasks. PI is only an approximation of the persistence diagram. The L2 distance between PIs does not accurately reflect the true Wasserstein distance between diagrams. Therefore, using an L2-distance-based loss to directly learn the PI may lead to the loss of important structural information carried by a diagram. An example is provided in Figure 5 in the revised supplementary material. For a sample vicinity graph from Cora, we compare the ground truth PI (computed from the ground truth diagram), the PI computed from the diagram estimated by our method PDGNN, and the PI directly estimated by GIN.  Both estimated PIs have similar L2 distance from the ground truth PI. But we observe that the PI estimated by PDGNN has a very similar spatial distribution to the ground truth PI. This structural property, however, is not preserved by the directly estimated PI. Such loss of structural information of directly estimated PIs, although not captured by the L2 error, partially explains their worse representation power. In Table 2, the directly estimated PIs (PEGN(GIN_PI) and TLC-GNN(GIN_PI)) perform worse in the downstream task.
> This choice of estimating diagrams instead of PIs is a part of the overarching theme of our paper. As pointed out by Reviewer 8ijg, the main contribution of our paper is to transfer a complicated and uncontrollable learning process to a controllable process with algorithmic insight. This general principle also applies to our learning algorithm. We decompose the diagram computation algorithm into sub-algorithms, which can be approximated well by a GNN. This is the reason PDGNN approximates the diagrams much better than other baselines in Table 1.
>
> **(3) The performance of PDGNN on large sparse graphs.**
>
> In the experiments in the paper, the input is the k-hop vicinity graphs. On citation graphs, the vicinity graph remains small. On these small graphs, the exact computation algorithm like Gudhi has less overhead, and thus is unsurprisingly faster. We will clarify this in the paper.
> Indeed, on large and sparse graphs, our method outperforms strong baselines like Gudhi significantly. In the table below, we compare the running time (in seconds) on popular citation networks including Cora, Citeceer and PubMed. For each graph, we run experiments on the largest connected subgraph. We also report the number of nodes/edges of the selected subgraph.
>
>
> | |  Cora | Citeseer| PubMed |
> | :----:| :----: | :----: | :----: |
> |Node/Edge| 2485/5069 | 2120/3679 | 19717/44324 |
> | Fast [51] | 0.184 | 0.068 | 1.816 |
> | Gudhi [41] | 0.045 | 0.023 | 1.696 |
> |Ours | **0.006** | **0.005** | **0.007** |

---

> > ### Author Response · Authors · 2022-08-02
> > **Additional Response to Reviewer 5254**
> >
> > **(4) Experiment on other choices of filter functions.**
> >
> > Below we set centrality and clustering coefficient as the filter function, follow the settings in Table 1, and report the approximation error on Cora, Citeseer, and PubMed.
> >
> > | Data | | Cora |  | Citeseer | | PubMed|
> > | :----: |:----: |:----: |:----: |:----: |:----: |:----: |
> > | Eval | $W_2$  | PIE | $W_2$ | PIE | $W_2$ | PIE|
> > |Clustering |0.392 |1.53e-3 |0.178 |7.17e-4 |0.267 | 2.5e-3|
> > |Centrality | 0.332 | 4.14e-4 | 0.237 | 4.65e-4 | 0.322 | 3.75e-4|
> >
> > We also report computation time following the settings in Table 4. The only difference is that below we report the time to generate all vicinity graphs (rather than 1000 graphs as in Table 4).
> >
> > | Data | | Cora |  | Citeseer | | PubMed|
> > | :----: |:----: |:----: |:----: |:----: |:----: |:----: |
> > |Filter | Clustering | Centrality | Clustering | Centrality | Clustering |Centrality|
> > | Fast [51] | 2.10 | 2.21 | 1.16 | 1.31 | 38.07 | 39.05 |
> > | Gudhi [41] | 0.98 | 1.00 | 0.59 | 0.63 | 16.79 |16.24 |
> > | Ours | 11.25 | 11.30 | 13.61 | 13.62 | 66.19 | 67.29 |
> >
> > **Observations.** We observe (1) filter function only has minor influence on inference/computation speed, for both the computation algorithm and ours; (2) filter function does influence the approximation error. The reason is that different filter functions have different ranges; functions with larger ranges tend to have larger approximation error especially on PIE. This is another evidence that the distance function on PIs is not very robust for learning (regarding Q2 above).
> >
> >
> > **(5) EPD decomposition extended to simplicial complexes.**
> >
> > The proposed algorithm is restricted to 0D/1D EPD computation. It does not generalize to high-dimensional simplices.
> >
> > **(6) How is the value of k (k-hop) chosen?**
> >
> > The choice of k follows the settings of existing state-of-the-art works [51, 57]. In [57], Zhao et al. have already evaluated the influence of k, and selected the best k based on the empirical performance of the learning task.
> >
> > **(7) Experiments on the threshold value of average N/E or degree to decide which method is the fastest to compute/estimate the EPD.**
> >
> > To find the threshold, we use the well-known Stochastic Block Model (SBM) to generate synthetic graphs. We set the number of nodes in these synthetic graphs from 200 to 300, with 10 as the step. In these graphs, we randomly generate 5 different clusters, and set the probability of edges intra-cluster to 0.4, and the probability of edges inter-cluster to 0.1. In this way, we can obtain 11 graphs with different nodes and edges. We set node degree as the filter function, and add experiments on the largest connected components of these 11 graphs. The information of the selected connected graphs and the running time (second) are listed below. As shown in the Table, the threshold is around 100 nodes / 820 edges.
> >  | Node | 80 | 84 | 88 | 92 | 96 | 100 | 104 | 108 | 112 | 116 |120 |
> > | :----: |:----: |:----: |:----: |:----: |:----: |:----: |:----: |:----: |:----: |:----: |:----: |
> >  | Edge | 515 | 585 | 660 | 713 | 759 | 820 | 943 | 1012 | 1060 | 1152 |1231 |
> >  | Fast [51] | 6.8e-3 | 8.0e-3 | 8.9e-3 | 9.6e-3 | 1.0e-2 | 1.1e-2 | 1.3e-2 | 1.4e-2 | 1.4e-2 | 1.6e-2 | 1.7e-2|
> > | Gudhi [41] | **2.5e-3** | **3.2e-3** | **3.6e-3** | **3.9e-3** | **4.0e-3** | 4.8e-3 | 5.5e-3 | 6.0e-3 | 6.4e-3 | 6.6e-3 |6.8e-3|
> >  | Ours | 4.5e-3 | 4.5e-3 | 4.6e-3 | 4.6e-3 | 4.6e-3 | **4.6e-3** | **4.7e-3** | **4.7e-3** | **4.7e-3** | **4.7e-3** |**4.8e-3** |
> >
> > We then add experiments to evaluate the influence of density. We fix the node number of the SBM model to 250, and set the probability of edges intra-cluster from 0.5 to 0.7, and the probability of edges inter-cluster from 0.05 to 0.15. The steps for intra-cluster and inter-cluster are 0.02 and 0.01, respectively. In this way, we can obtain 11 graphs with the same nodes and different edges. We set node degree as the filter function, and add experiments on the largest connected components of these 11 graphs. The information of the selected connected graphs and the running time (second) are listed below. As shown in the Table, the threshold is around 100 nodes / 766 edges.
> >
> >  |Node | 100 | 100 | 100 | 100 | 100 | 100 | 100 | 100 | 100 | 100 | 100 |
> > | :----: |:----: |:----: |:----: |:----: |:----: |:----: |:----: |:----: |:----: |:----: |:----: |
> > | Edge | 489 | 529 | 595 | 652 | 766 | 842 | 968 | 1011 | 1082 | 1231 |1307 |
> > | Fast [51] | 7.0e-3 | 7.4e-3 | 8.3e-3 | 9.0e-3 | 1.1e-2 | 1.2e-2 | 1.3e-2 | 1.4e-2 | 1.4e-2 | 1.6e-2 |1.7e-2|
> > | Gudhi [41] | **2.8e-3** | **2.9e-3** | **3.0e-3** | **4.1e-3** | **4.2e-3** | 5.1e-3 | 5.5e-3 | 5.9e-3 | 6.2e-3 | 6.3e-3 | 6.7e-3 |
> > | Ours | 4.1e-3 | 4.1e-3 | 4.2e-3 | 4.2e-3 | 4.3e-3 | **4.4e-3** | **4.7e-3** | **4.7e-3** | **4.8e-3** | **4.8e-3** | **4.8e-3** |

---

> > > ### Author Response · Authors · 2022-08-02
> > > **Additional Response to Reviewer 5254**
> > >
> > > **(8) The influence of k (k-hop) in efficiency.**
> > >
> > > We evaluate the efficiency on vicinity graphs with different k’s. In Table 4, we already reported the results when k=2. Here we report the results when k=1 and k=3. For k=1, following the same settings as Table 4, we report the statistics of the vicinity graphs, and the time to generate/estimate EPDs below.
> > >
> > > |Dataset | Cora | Citeseer | PubMed | Photo | Computers | CS | Physics |
> > > | :----: |:----: |:----: |:----: |:----: |:----: |:----: |:----: |
> > > | Avg. N/E | 4.9/10.6 | 3.8/7.7 | 5.6/11.7 | 31.1/333.6 | 38.7/430.7 | 9.6/32.0 | 13.1/53.2 |
> > > | Fast [51] | 0.194 | 0.097 | 0.115 | 4.083 |5.827 | 0.313 |0.713|
> > > | Gudhi [41] | **0.088** | **0.074** | **0.082** | **2.387** | 5.249 | **0.179** | **0.279**|
> > > | Ours | 4.086 | 4.012 | 4.073 | 4.087 | **4.097** | 4.085 | 4.095 |
> > >
> > > For k = 3, we set the number of vicinity graphs to 100 (originally 1000), and report the results as below.
> > >
> > > |Dataset | Cora | Citeseer | PubMed | Photo | Computers | CS | Physics |
> > > | :----: |:----: |:----: |:----: |:----: |:----: |:----: |:----: |
> > > | Avg. N/E | 131.1/371.7 | 53.4/163.8 | 363.2/1380.2 | 2465.0/45497.5 | 8549.3/191947.6 | 813.1/4378.8 | 2028.9/18638.2 |
> > > | Fast [51] | 0.38 | 0.14 | 2.13 | 102.27 | 473.71 | 8.36 | 45.77 |
> > > | Gudhi [41] | **0.14** | **0.07** | 1.26 | 219.19 | 2499.24 | 8.09 | 137.83 |
> > >  | Ours | 0.43 | 0.42 | **0.44** |  **0.62** | **1.21** | **0.47**| **0.54**|
> > >
> > > **Observation.** we can observe the same tendency as in Table 4 in the main paper. The performance is generally correlated with the size/density of the vicinity graphs. For k=1, for all benchmarks, the vicinity graph is small. Thus sequential algorithms are generally faster. For k=3, the vicinity graph is very large. Our method is significantly faster, especially on Photo, Computers, CS, and Physics.

---

> ### Author Response · Authors · 2022-08-05
> **Soliciting further questions**
>
> Dear Reviewer,
>
> We have tried our best to provide additional empirical results to address your concerns. Please do not hesitate to let us know if you have any further questions. We will be very happy to answer them during the discussion period.
>
> Sincerely,
> Authors

---

> > ### Comment · Reviewer_5254 · 2022-08-09
> > **Thanks for your response**
> >
> > Thank you for the detailed response. I would like to revise my rating as most of my questions are answered. A quick follow-up question: which kind of centrality did you use?

---

> > > ### Author Response · Authors · 2022-08-09
> > > **Thanks for your support**
> > >
> > > Thanks very much for your support!
> > >
> > > We use the degree centrality function implemented with the package "networkx==2.5". The algorithm is available in https://networkx.org/documentation/stable/_modules/networkx/algorithms/centrality/degree_alg.html#degree_centrality.  And we will clarify this in the paper.

---

### Meta-Review · Area_Chair_KC2k · 2022-08-26

**Recommendation:** Accept
**Confidence:** Certain

**Metareview:**

A reasonably interesting paper on deep models to approximate a popular class of objects, extended persistence diagrams, in topological data analysis. This is both an important problem, since these can be used throughout various parts of graph machine learning, and is a challenging one that requires technical innovations. The authors deliver on this, introducing quite good results.

The reviewers (and I) are in agreement about the paper, and the additional results the authors provided in response to the reviews are convincing. For example, on large sparse graphs, the proposed algorithms maintain a big speedup advantage.

Perhaps the only question that came up is whether this work is more of a TDA-specific topic, but in general the paper clearly fits well within the machine learning community.

**Award:**

No

---

### Decision · Program_Chairs · 2022-09-14

Accept